# Deciphering infected cell types, hub gene networks and cell-cell communication in infectious bronchitis virus via single-cell RNA sequencing

**Chengyin Liukang**[1,2☯], **Jing Zhao**[1,2☯], **Jiaxin Tian**[1], **Min Huang**[1], **Rong Liang**[1], **Ye Zhao**[1,2],
**Guozhong Zhang**[1,2]*

1 National Key Laboratory of Veterinary Public Health Security, College of Veterinary Medicine, China Agricultural University, Beijing, People's Republic of China, 2 Key Laboratory of Animal Epidemiology of the Ministry of Agriculture, College of Veterinary Medicine, China Agricultural University, Beijing, China

☯ These authors contributed equally to this work.
* zhanggz@cau.edu.cn

**Data Availability Statement:** All relevant data are provided as figures within the paper and its supporting information files. Sequencing data

## Abstract

Infectious bronchitis virus (IBV) is a coronavirus that infects chickens, which exhibits a broad tropism for epithelial cells, infecting the tracheal mucosal epithelium, intestinal mucosal epithelium, and renal tubular epithelial cells. Utilizing single-cell RNA sequencing (scRNA-seq), we systematically examined cells in renal, bursal, and tracheal tissues following IBV infection and identified tissue-specific molecular markers expressed in distinct cell types. We evaluated the expression of viral RNA in diverse cellular populations and subsequently ascertained that distal tubules and collecting ducts within the kidney, bursal mucosal epithelial cells, and follicle-associated epithelial cells exhibit susceptibility to IBV infection through immunofluorescence. Furthermore, our findings revealed an upregulation in the transcription of proinflammatory cytokines IL18 and IL1B in renal macrophages as well as increased expression of apoptosis-related gene STAT in distal tubules and collecting duct cells upon IBV infection leading to renal damage. Cell-to-cell communication unveiled potential interactions between diverse cell types, as well as upregulated signaling pathways and key sender-receiver cell populations after IBV infection. Integrating single-cell data from all tissues, we applied weighted gene co-expression network analysis (WGCNA) to identify gene modules that are specifically expressed in different cell populations. Based on the WGCNA results, we identified seven immune-related gene modules and determined the differential expression pattern of module genes, as well as the hub genes within these modules. Our comprehensive data provides valuable insights into the pathogenesis of IBV as well as avian antiviral immunology.

have been deposited in Genome Sequence Archive (GSA) of National Genomics Data Center under the accession number CRA015620 (https://bigd.big.ac.cn/gsa/browse/CRA015620).

**Funding:** This study was supported by the National Key Research and Development Program of China (2021YFD1801103) and the 2115 Talent Development Program of China Agricultural University to GZ. The funders had no role in study design, data collection and analysis, decision to publish, or preparation of the manuscript.

## Author summary

A comprehensive understanding of the pathogenesis of IBV in chickens is crucial for the development of effective control and prevention strategies. In our study, we conducted single-cell RNA sequencing analyses on chicken kidney, bursa, and tracheal tissues to systematically uncover the cell types targeted by IBV infection and to delineate the immune response patterns and key genes within immune gene modules in different tissues. Consequently, this research holds the potential to provide novel targets and directions for the prevention of IBV.

## Introduction

Infectious bronchitis (IB) is a significant avian disease caused by the infectious bronchitis virus (IBV), posing a substantial threat to the global poultry industry [1]. IBV belongs to the γ-coronavirus genus of the Coronaviridae family. The disease was first observed in North Dakota, USA, in 1930, with the first publication on IB following in the subsequent year [2]. The diversity of IBV strains is attributed to their inherent high mutation rate and frequent recombination events, which are common among coronaviruses [3]. Mutations primarily arise from errors during transcription process by the virus's RNA-dependent RNA polymerase (RdRp), while recombination occurs due to RdRp stuttering or slippage [4]. Transmission of IBV predominantly occurs through aerosols, with intermittent shedding by infected chickens contaminating the environment and serving as a source of infection for susceptible birds. Clinical manifestations of IBV infection include respiratory symptoms such as coughing, sneezing, rales, and nasal and ocular discharges. The virus can infect and induce a rapid apoptosis of ciliated cells lining the trachea, thereby increasing susceptibility to other pathogens [2]. Certain strains of IBV exhibit nephropathogenicity leading to renal urate deposition and severe nephritis. These nephropathogenic strains may result in high mortality rates [5]. Additionally, IBV infection can cause damage to laying hens' oviducts resulting in a significant reduction in egg production by up to 70% in severe cases [6,7]. Therefore, comprehending interactions between IBV and its host cells is crucial for developing successful preventative and therapeutic strategies.

Severe nephritis resulting from renal infection is the primary cause of mortality in IBV infection. Previous studies have demonstrated that IBV can replicate in renal tubular epithelial cells and induce severe kidney lesions [8]. Transcriptomic analysis of IBV-infected kidneys has revealed upregulation of innate immune genes, such as Toll-like receptors, with IBV also promoting cytokine release through the TLR7/NF-κB signaling pathway, thereby contributing to renal inflammation [9–11]. Moreover, IBV infection is associated with a decline in renal antioxidant capacity and metabolic dysfunction [12]. However, due to the diverse cell types present in the kidney and distinct immune and inflammatory responses activated post-infection, investigations based on whole tissue samples have limitations in elucidating the pathogenicity of IBV. In addition to respiratory system and kidney infections, certain strains of IBV can also infect the bursa of Fabricius, leading to pathological damage and potential immunosuppressive risks [13,14]. With advancements in single-cell sequencing technology, high-throughput transcriptomics at single-cell resolution has become feasible. Single-cell mRNA expression data enables a more comprehensive understanding of the specific cell types involved in viral infection as well as the immune responses elicited post-infection across different cell types [15–17].

In this study, we generated a comprehensive single-cell atlas of IBV-infected tracheal, renal, and bursal tissues using the microfluidic 10x Genomics scRNA-seq platform. We analyzed

samples from eight specific-pathogen-free (SPF) chickens to obtain single-cell expression data. Initially, we identified specific cell markers for various cell types and conducted an analysis of genes exhibiting robust expression specific to each cell type. Subsequently, we evaluated changes in the proportions of each cell type and potential target cells after IBV infection. To validate the target cells of IBV infection, immunofluorescence was performed revealing that tracheal mucosal epithelial cells, renal distal tubules and collecting duct cells, bursal mucosal epithelial cells, and follicle-associated epithelial cells can be infected by IBV. Furthermore, utilizing the R package CellChat, we investigated potential intercellular interactions in different tissues post-IBV infection. Finally, weighted gene co-expression network analysis (WGCNA) was applied to integrate single-cell data from multiple tissues. This analysis identified several immune-related gene modules and their expression patterns across different cell populations and samples after IBV infection. Additionally, hub genes within these immune gene modules were also discovered by network analysis. These findings contribute to our understanding of host-IBV interactions while providing novel insights into avian antiviral immunology.

## Results

### Tissue tropism of IBV in chickens

To investigate the tissue damage caused by IBV infection and to preliminarily identify the cell types infected, we collected kidney, bursa, and trachea tissues at 5 days post-inoculation (dpi) with IBV. Tissue samples were fixed using paraformaldehyde and processed into tissue sections for both hematoxylin and eosin (H&E) staining as well as immunofluorescence. After IBV infection, severe inflammatory responses were observed in the kidneys, characterized by extensive infiltration of inflammatory cells, necrosis of tubular cells, and occlusion (Fig 1A). Immunofluorescence revealed that renal tubular cells, bursal mucosal epithelial cells, and tracheal epithelial cells were susceptible to IBV infection (Fig 1A and 1B). Furthermore, our observations demonstrated that IBV could also infect the epithelial cells of the Harderian gland and the mucosa of cecal tonsils (S1 Fig). Importantly, viral positive staining was not detected in tissues other than renal tubules and mucosal epithelium, suggesting a tropism of IBV towards mucosal epithelial cells.

### Identification of cell clusters in the kidney tissues and cell types infected by IBV

To investigate the infection status and immune response of different cell types in the kidney after IBV infection, we collected kidneys from infected and control groups at 5 dpi. Single-cell sequencing was performed using the 10x Genomics platform, resulting in a comprehensive single-cell transcriptomic profile of the kidney. After data quality control, removing cells with high mitochondrial gene expression and doublets, we obtained a total of 14,644 cells (7,040 from the infected group and 7,604 from the control group). Following exclusion of erythrocytes expressing high levels of HBBA and RFESD, expression profiles for 13,473 cells were acquired. Using uniform manifold approximation and projection (UMAP) for dimensionality reduction, we identified 18 cell clusters (S2A Fig). Cell types were characterized based on marker gene expression patterns: LRP2$^+$ proximal tubular cells (cluster 0, 1, 3, 4, 5, 7, 8, 10, 11), CALB1$^+$ distal tubular cells along with AQP2$^+$ collecting duct cells clustered together (cluster 12), ATP6V0A4$^+$ intercalated cells (cluster 14, 18), CD3D$^+$ CD4$^+$ CD4 T-cells (cluster 2), CD3D$^+$ CD8A$^+$ CD8 T-cells (cluster 9, 6, 15), CD3D$^+$ KK34$^+$ γδ T-cells(cluster 13), and CSF1R$^+$ macrophages(clusters 16, 19) (Fig 2A and 2B). Due to reptilian nephrons predominating in chicken kidneys where proximal tubules directly connect to distal tubule cells, Henle's loop cells were not identified in our single-cell data. We also identified genes specifically expressed in each cell type (Fig 2C). Compared to the control group, significant changes in the

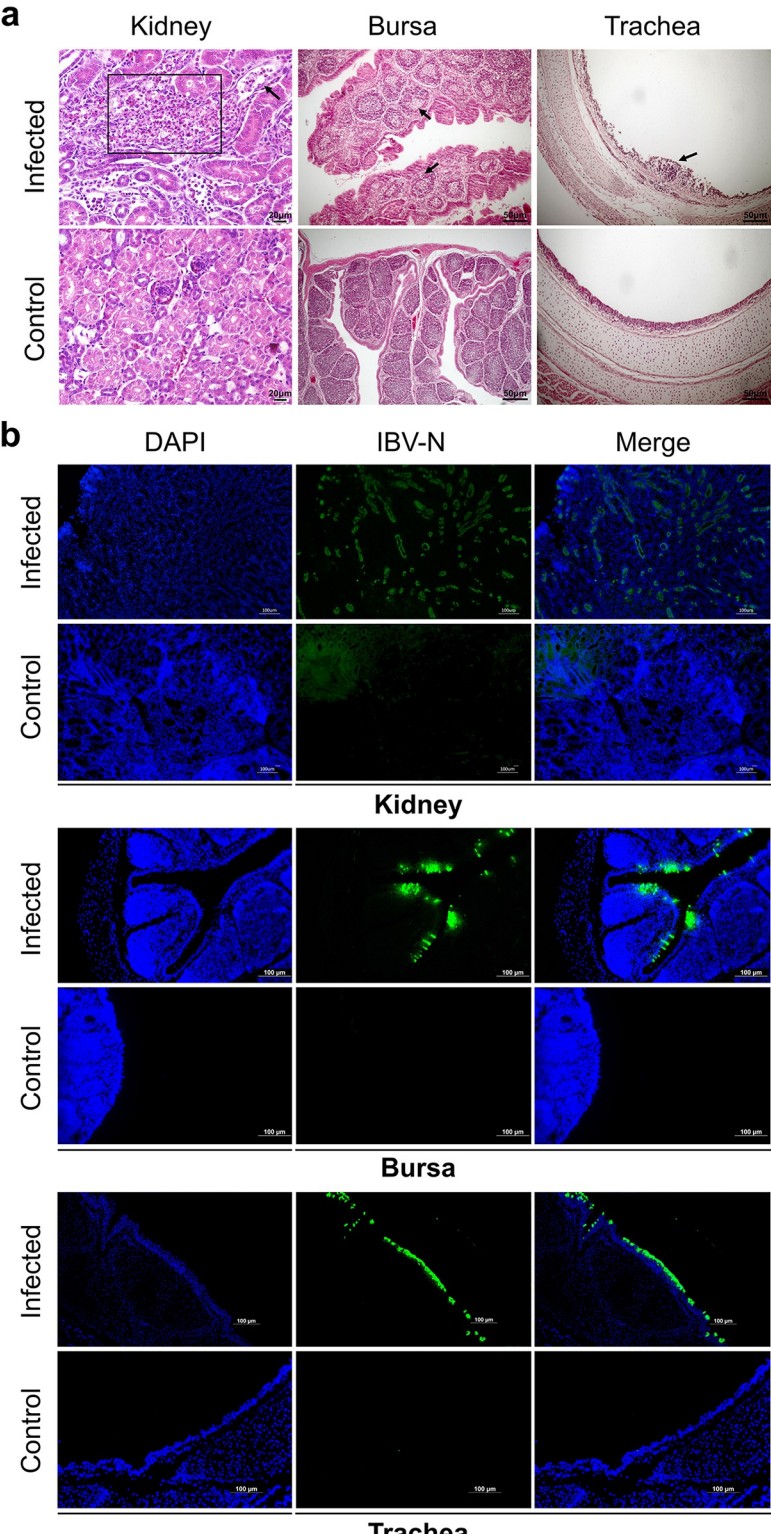

**Fig 1. Histopathological staining and immunofluorescence analysis.** (a) Histopathological staining of tissues. In the kidney section, the black square highlights immune cell infiltration, while the black arrows indicate tubular cell casts and obstructions. The bursa of Fabricius section shows damage and depletion of lymphoid follicles, as indicated by black arrows. In the trachea section, arrows point to immune cell infiltration. (b) Immunofluorescence using IBV-N antibody in kidney, trachea, and Bursa tissues. Green fluorescence indicates viral positivity.

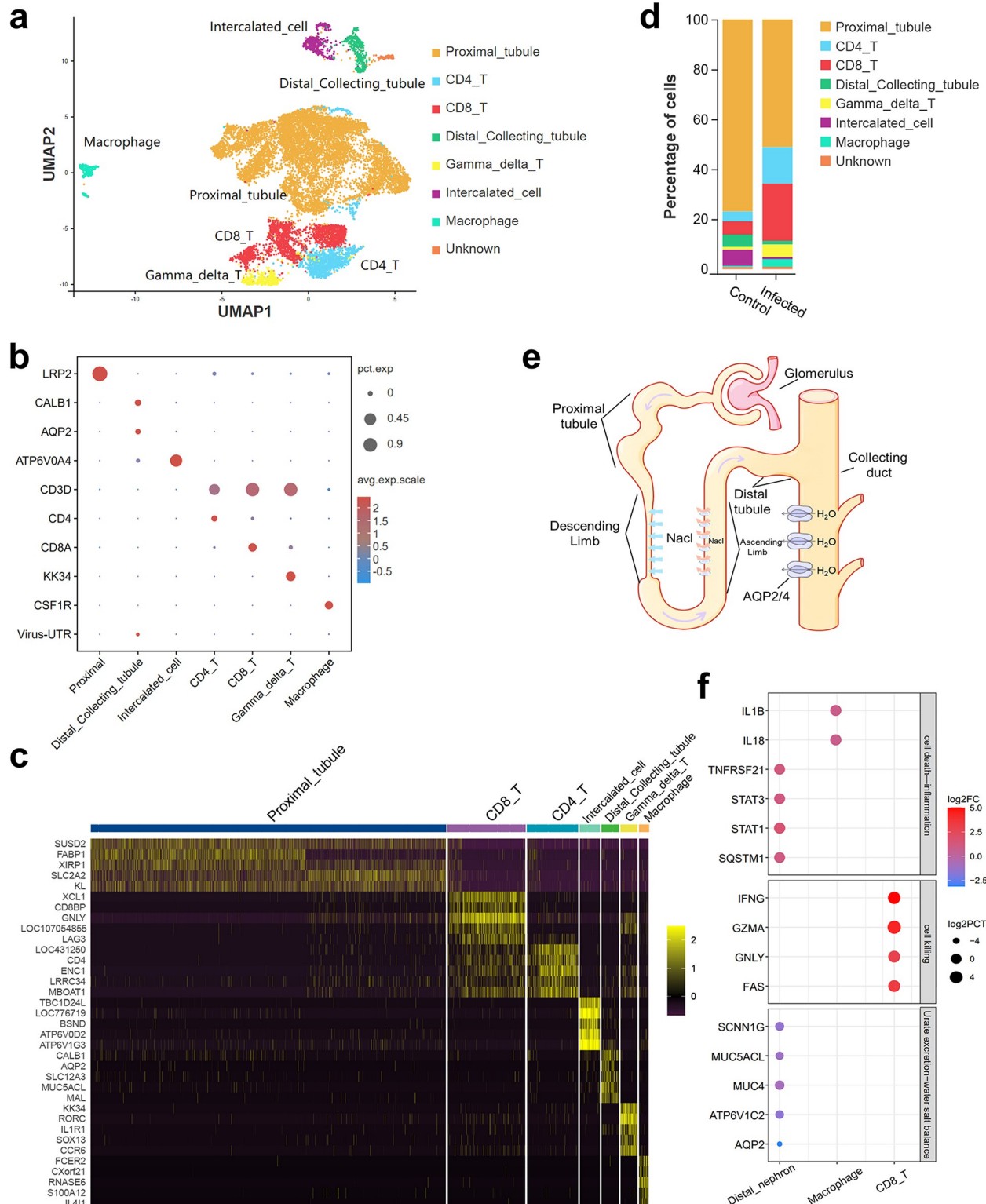

**Fig 2. Single-cell profiling of cell populations in the kidney collected from IBV infected and control chickens.** (a) UMAP projection representing the cell clusters identified by specific molecular markers. (b) Cell type annotation and dot plot representing characteristic genes (row) in each cluster (column). Dot size represents the proportion of cells in the cluster that express the gene; intensity indicates the expression level (Z-score) relative to those from other clusters. (c) Heatmap depicting the expression levels (Z-scores) of signature genes across each cluster. (d) Proportion of cell types in renal tissue before and after infection. (e) Components and structure of the nephron. (f) Expression of inflammatory

genes and salt excretion genes in immune cells and distal nephron cells. The dot size represents the change in the percentage of gene-expressing cells (infect group cell percentage / control group cell percentage), while the color reflects changes in gene expression levels (infected group average expression / control group average expression).

proportions of various cell types were observed after IBV infection. There was an increase in the percentages of T cells (from 10.43% to 42.5%) and macrophages (from 0.56% to 3.06%). Conversely, there was a decrease in the components of the distal renal unit, including distal tubules and collecting ducts (4.88% - 1.51%), as well as intercalated cells (6.43% - 0.84%) (Fig 2D).

The 10x Genomics platform selectively captures polyadenylated mRNA and sequences approximately 150bp upstream of the mRNA polyA tail, which is located in the 3' UTR in coronavirus. We identified the cell types infected by the virus based on viral 3' UTR expression. Notably, distal tubules and collecting duct cells exhibited the highest levels of viral RNA expression (Fig 2B and S1 Table). In order to validate the findings in the single cell sequencing data, we collected kidney tissues from IBV-infected chickens at 5 dpi and performed dual immunofluorescence using IBV N protein antibody along with collecting duct specific marker AQP2 or distal tubule specific marker CALB1. AQP2 is specifically localized to the apical plasma membrane of the collecting duct, while the IBV N protein is localized in the cytoplasm. In the merged images, the IBV N protein can be observed in cytoplasm of AQP2-expressing collecting duct cells. Similarly, CALB1 localizes with the IBV N protein in the same distal tubule cells (Fig 3A and 3B). Our observations confirmed that IBV could infect the kidney collecting duct and distal tubule cells. These results corroborate the single-cell sequencing data.

## Impact of IBV infection on gene expression and inflammatory response in chicken kidney tissue

To investigate the molecular mechanism of renal tissue injury caused by viral infection, we utilized the R package Seurat to perform differential gene expression analysis on distal nephron cells (including collecting duct and distal tubule), $CD8^+$ T-cells, and macrophages between IBV-infected and control groups (S2 Table). Additionally, enrichment analysis was conducted for the identified differentially expressed genes. Our findings revealed a significant up-regulation of immune genes and necrosis-related genes in the distal nephron of IBV infected group. Conversely, genes responsible for water and salt metabolism, ion transport, and urate excretion were down-regulated. Furthermore, there was an up-regulation of pro-inflammatory cytokines IL1B and IL18 in macrophages and CD8 T cells exhibited increased expression of cytotoxic and cell-killing genes (Figs 2F, S3 and S4). Notably, the expression of mucin (MUC4) in the collecting duct, which plays a pivotal role in urate excretion [18–20], exhibited significant down-regulation. Moreover, AQP2 and SCNN1G involved in water reabsorption and $Na^+$ reabsorption [20,21] were also significantly down-regulated (Figs 2F and S4). Our study demonstrated that IBV infection induced upregulated expression of inflammatory genes in the distal nephron, while disrupting water-salt balance and expression of genes related to urate metabolism, which may be the cause of kidney inflammation and large amount of urate deposition induced by IBV infection in chickens. These combined factors contribute to severe renal urate accumulation accompanied by swelling and inflammation ultimately resulting in chicken mortality [22].

## Changes of cell-cell communication in chicken kidney after IBV infection

Cell-cell communication analysis in single-cell sequencing data computationally infers signaling interactions or intercellular crosstalk, elucidating how cells communicate and coordinate

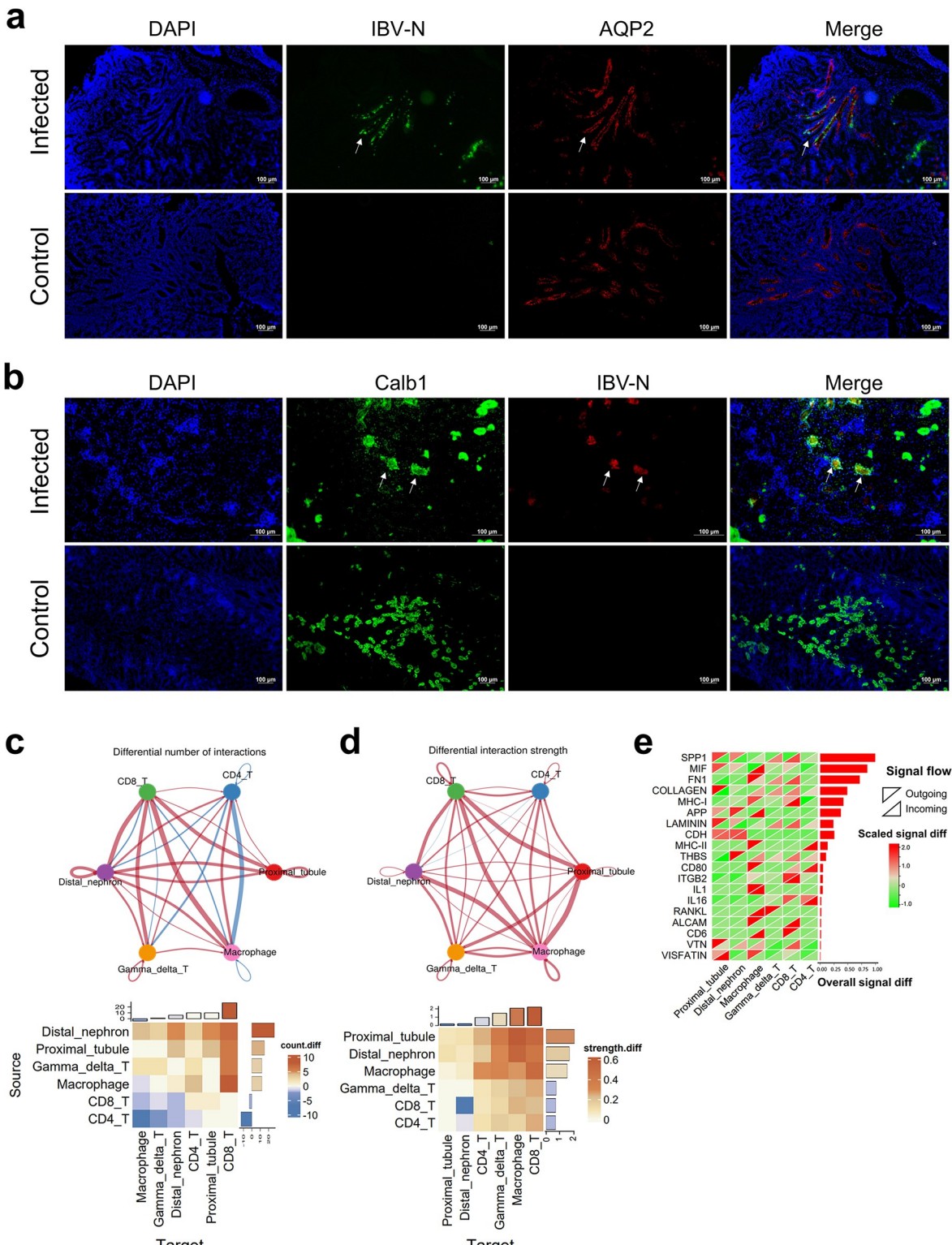

**Fig 3. Dual immunofluorescence identifying cell types infected by IBV in chicken kidney and changes in cell-cell communication after IBV infection.** (a) White arrows indicate the localization of IBV N protein in AQP2-expressing collecting duct cells. Green fluorescence shows positive staining for IBV N and red fluorescence shows staining for AQP2. (b) White arrows indicate the localization of IBV N protein in CALB1-expressing distal tubule cells. Red fluorescence shows positive staining for IBV N and green fluorescence shows staining for CALB1. (c) Changes in cell communication numbers: The top network diagram shows cell clusters as nodes, with line

thickness indicating changes in communication numbers. The lower heatmap details these changes, with rows representing signal-sending cells and columns indicating signal-receiving cells. The color scale reflects the inter-group differences in signal communication frequency between different cell types (number of communications in the infected group—number in control group). The bar plots at the top and right side represent the overall differences in the number of signals sent/received by specific cell clusters. (d) Changes in cell communication strength: Similar to (c), with the top network diagram displaying changes in communication strength (communication strength in the infected group—strength in the control group). In the lower heatmap, the color scale reflects the inter-group differences in signal communication strength between different cell types. The bar plots at the top and right side represent the overall differences in the strength of signals sent/received by specific cell clusters (infected group—control group). (e) Inter-group differences in the communication strength of specific signaling pathways (receptor-ligand pairs) across cell clusters. Rows represent signal pathways and columns correspond to cell clusters, with heatmap colors depicting the strength variation of signals (infect group vs. control group). Upper left triangles for signal sent and lower right triangles for signal received. The bar plot on the right side shows the overall difference in communication strength of these signaling pathways between the IBV-infected group and the control group.

through secreted signaling molecules. Two main methods exist: 1) The ligand-receptor method, based solely on the expression patterns of ligand genes in signaling cells and receptor genes in receiving cells, exemplified by algorithms like CellChat [23]. 2) The ligand-receptor target method, which additionally incorporates the expression of downstream signaling pathway genes in receiving cells. While the latter considers downstream targets of ligands, it is less robust due to the complex regulation and crosstalk of intracellular signaling pathways, and its reliance on the accuracy of existing pathway databases like MultinicheNet [24]. These two analytical methods can complement each other, providing a more comprehensive landscape of cell-cell communication [25,26].

To investigate the interaction changes between different cell types in the kidney after IBV infection, we first utilized R package Cell-chat and the ligand-receptor pairing information from CellchatDB databases to assess cellular communication based on single-cell gene expression data and cell grouping information. The communication type and strength between cells were determined by evaluating the expression levels of ligand-receptor pairs, with permutation tests used to calculate the p-value for each pairs communication probability. Only ligand-receptor pairs with a p-value < 0.05 were retained (S3, S4, and S5 Tables). Initially, we evaluated changes in cell-to-cell communication after IBV infection, observing that distal nephron and proximal tubule cells served as primary signal source, exhibiting significantly increased numbers and strength of signals post-infection. Conversely, CD8[+] T cells and other immune cells acted as main signal receivers, displaying increased numbers and strength of received signals (Fig 3C and 3D). Macrophages served as signal receivers; the number of signals decreased but the strength increased significantly after IBV infection. We further investigated changes in specific signaling pathways (Fig 3E). Amongst up-regulated pathways after IBV infection, proximal tubular and distal nephron cells serve as key signal senders associated with secreted phosphoprotein 1 (SPP1), migration inhibitory factor (MIF), amyloidogenic pathway (APP), thrombospondin pathway (THBS), and cadherin pathway activation. The receiver cells primarily consisted of immune cells. In macrophages, the MIF, APP, IL1, and Receptor activator of NF-κB ligand (RANKL) signaling pathways exhibited significant enhancement. Importantly, both the secreting and receiving cells involved in the IL1 pathway were macrophages, indicating their self-activating role in the inflammatory response. Furthermore, within the macrophage signaling pathways, fibronectin (FN1), MHCI, MHCII, CD80 were significantly upregulated; these pathways predominantly interacted with T cells during migration, highlighting the crucial role of macrophage in inflammation and immune regulation. The communication patterns of CD8[+] T cells and γδT cells demonstrated a high congruence as receiver in various signaling pathways including SPP1, FN1, MHCI, THBS, and integrin subunit beta 2 (ITGB2). The γδT cells as source in the RANKL pathway suggests their contribution to macrophage activation. CD4[+] T cells primarily served as signal receivers responding to MHCII signals and Interleukin 16 (IL16) signals (Fig 3E). In summary, our cell

communication data revealed an overall pattern of interactions among different cell types following viral infection while identifying specific ligand-receptor pairs that play essential roles in each pathway.

Building upon this, we employed MultiNicheNet for cell-cell communication analysis. Our findings indicate that, within kidney samples from the infected group, IL1B is predominantly secreted by macrophages, aligning with predictions made by CellChat. Beyond macrophages, IL1B also targets CD8 T cells. In mammals, IL21 signaling is primarily initiated by CD4 T cells; however, our data reveal that in this context, IL-21 is chiefly produced by CD8 T cells, influencing distal and collecting duct cells, CD4 T cells, and CD8 T cells. This might reflect unique aspects of the avian immune system. The critical antiviral immune response mediator, IFNγ (IFNG), is produced by CD8 T cells and received by both macrophages and CD8 T cells, highlighting the potential role of avian CD8 T cells in promoting self-activation and the activation of macrophages. Furthermore, macrophages can secrete CCL1 signals, which play a role in recruiting CD8 T cells through the CCL1-CCR8 receptor interaction (S6A Fig and S6 Table).

Subsequently, we utilized the MultiNicheNet algorithm to construct a ligand-target gene regulatory network among cells. Briefly, MultiNicheNet integrates database and experimental data to establish a ligand-target gene regulatory network. Then, by examining the gene expression patterns between different groups (in this case, infected vs. control groups) and the correlation in ligand-target gene expression, it predicts differential regulatory patterns among various cell clusters. We discovered that signals from multiple cell types can activate CD8 T cells to express IFNγ in the infected group, including IL15 and SPP1 signals from proximal tubule cells and CD28, CD45 (PTPRC) signals from CD4 T cells. CD40LG, which plays a crucial role in antigen-presenting cell activation, is predominantly expressed by CD4 T cells. It is activated by signals from CD28 of CD8 T cells and SPP1 of renal tubule cells, leading to the activation of genes such as CD38 and selectin ligand (SELPLG). Distal-collecting duct cells receive signals from macrophages and CD4 T cells, like IL1B and CD44, express IL-6, and sequentially activate γδT cells to express IL17, playing a potential role in T cell activation and the recruitment of neutrophils (S6B Fig).

In summary, through two representative methods of cellular communication analysis, we have obtained a comprehensive picture of intercellular signaling and regulatory interactions within viral-infected kidney tissue.

## Identification of cell clusters in chicken bursa and cell populations infected by IBV

To investigate the infection status and immune response in different cell types in the bursa, bursa tissues were collected from chickens at 5 dpi as well as control chickens. Single-cell sequencing was performed using 10×Genomics platform. We obtained a total of 12,707 cells for transcriptional profiling (6,335 cells from the infection group and 6,372 cells from the control group). By applying clustering UMAP dimensionality reduction, we identified 14 distinct cell clusters (S2B Fig). Based on gene expression patterns, we classified 7 different cell types (Fig 4A). B cells were identified by chicken B cell specific marker Bu-1 (cluster 0,1,2,3,6,9,10,13). Macrophages (cluster 8) and dendritic cells (cluster 12) were distinguished based on the expression of TIMD4 and CSF1R genes, respectively, which have been identified previously [27]. CD8 T cells (cluster 7) were characterized by the expression of CD3D and CD8A genes. The mucosa surface of bursa was covered by interfollicular epithelial cells (IFE, cluster 5 and cluster 11), which exhibited mucus secretion capabilities along with expression of mucin (MUC4) and keratin (KRT7). Reticular epithelial cells (ERC) in the follicular medulla

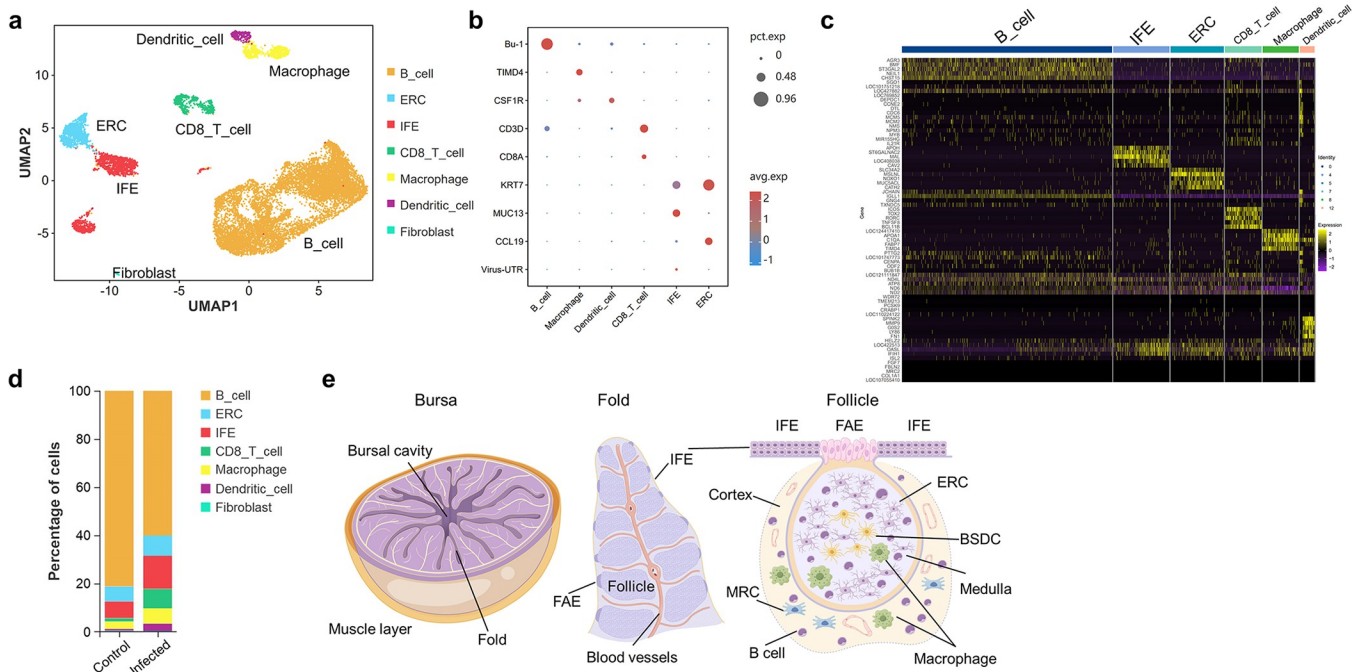

**Fig 4. Single-cell transcriptomic profiles of bursa tissues from IBV-infected and control chickens.** (a) UMAP dimensional reduction and clustering, cells are categorized into subpopulations based on molecular specificity markers. (b) Bubble plot illustrating the expression of cell type-specific molecular markers and viral RNA. The rows display gene names, and the columns represent cell populations. Bubble size indicates the proportion of gene expression in each cell type, while color denotes normalized gene expression levels (Z-score). (c) Heatmap depicting the expression levels (z-score) of characteristic genes in each cluster. (d) Bar graph showing the proportion of various cell types before and after infection. (e) Schematic representation of the Bursa of Fabricius structure. IFE, interfollicular epithelium; FAE, follicle associated epithelial cell; ERC, reticular epithelial cell; BSDC, bursa secretory dendritic cell.

formed a scaffold facilitating immune cell migration similar to mammalian lymph node follicular reticular fibrocytes (Fig 4E). Follicular reticular fibrocytes highly expressed chemokine CCL19 which plays a crucial role in dendritic cell and T-cell homing and homeostasis [28]. We found CCL19 also served as a specific molecular marker for identifying ERCs. In summary, we have identified 7 distinct cell types in the bursa based on the molecular markers (Fig 4A and 4B). The percentage of CD8 T-cells, macrophages and dendritic cells significantly increased after IBV infection, indicating the activation and recruitment of immune cells (Fig 4D). By evaluating the expression of viral 3'UTR RNA in different cell populations, we observed significant upregulation of viral 3'UTR RNA in both IFEs and macrophages, suggesting that these cell populations may serve as main target of IBV (Fig 2B and S1 Table).

## IBV has the capability to infect bursa inter-follicular epithelial cells and follicle-associated epithelial cells

To further characterize the specific cell types infected by IBV, we collected bursa tissues from chickens at 5 dpi and performed immunofluorescence on tissue sections using antibodies against the viral N protein. Initially, we observed that IBV specifically infects the monolayer cells lining the mucosal surface of the bursa (S1 Fig). Interestingly, IBV infection was confined to the mucosal epithelial region and didn't extend into the bursa follicles below the mucous membrane. The mucosal surface of bursa is lined by two distinct cell types: the interfollicular epithelial (IFE) cells that separate the bursal follicles and the follicle-associated epithelial (FAE) cells that cover the bursal follicles (Fig 4E). At the FAE sites, macrophages traffic between the bursal follicles and the lumen, taking up antigens and presenting them to the B lymphocytes

within the follicles. FAE cells also play a pivotal role in mucosal immunity through antigen uptake and transport to underlying immune cells. FAE cells specifically express colony-stimulating factor receptor (CSF1R). Obviously, IBV could infect IFE cells that separate the bursal follicles (S1 Fig). Next, we performed frozen section of bursal tissues from IBV-infected chickens and carried out dual immunofluorescence staining with CSF1R antibody along with IBV N protein antibody, demonstrating that IBV can infect bursal FAE cells (Fig 5A). Considering the detection of viral RNA within macrophages in single-cell sequencing data, we performed dual immunofluorescence on bursa using antibodies against TIMD4 and the IBV N protein to investigate whether IBV could infect TIMD4$^+$ macrophages located at FAE and bursa follicles. The results showed that that the IBV N protein was not located in macrophages expressing TIMD4 (Fig 5B). Our results indicate that IBV can infect bursal IFEs and FAEs but not infect macrophages.

## Changes of cell-cell communication in bursa after IBV infection

As in the previous cell-cell communication analysis in the kidney, to investigate the interaction among cell populations in bursa, we utilized the R package Cell-Chat and the CellchatDB database to evaluate intercellular communications, while retaining ligand-receptor pairs with p-value<0.05. We assessed changes in the number of cell-cell communication among different populations after infection and found that apart from B cells acting as signal sources and macrophages as signal receivers, there was no significant upregulation in cell-cell communications among other cell populations (Fig 5C and S7 Table). In contrast, there was a general upregulation in the strength of cellular communication among these populations. Dendritic cells and reticular epithelial cells served as both sources and receivers and exhibited significant upregulation both in signal emission and reception strength. B cells and IFEs showed a notable enhancement in signal emission strength while their signal reception strength remained unchanged. CD8 T-cells and macrophages, primarily serving as signal receivers, exhibited a significant enhancement in signal reception strength (Fig 5D and S7 Table). Based on the ligand-receptor pair data, we analyzed the changes in specific signaling pathways after infection. The Midkine pathway (MK) was notably upregulated, predominantly sent by B cells, while dendritic cells, ERC, and IFE mainly served as receivers. APP, CDH, MHCI, and Selectin E (SELE) pathways were primarily sent by epithelial cells, with APP signals received by B cells, dendritic cells, and macrophages, CDH pathway signals received by the epithelial cells themselves, MHCI signals received by CD8 T cells, and SELE pathway signals received by CD8 T and dendritic cells. The MIF signal was mainly sent by B cells and CD8 T cells, while it was received by macrophages and dendritic cells. IL1 signals were predominantly self-secreted and received by macrophages and dendritic cells. The integrin subunit beta 2 (ITGB2) pathway was primarily activated in B cells, CD8 T cells, and dendritic cells, with CD8 T cells serving as the receivers (Fig 5E and S8 and S9 Tables). Our cell interaction data revealed the overall interaction patterns among various cell types after viral infection and identifies the active ligand-receptor pairs within different pathways.

Next, utilizing the MultiNicheNet algorithm, we initially identified the top 40 cellular communication interactions in bursa samples from infected and control groups. In the IBV infected group, signals of IL6, IL1B, and IL15 were predominantly sent by dendritic cells, with receiver cells being macrophages, the dendritic cells themselves, intestinal follicle epithelium (IFEs), and CD8 T cells. Additionally, the CSF1 signal, also originating from dendritic cells, was received by macrophages and dendritic cells via CSF3R and CSF1R, playing a role in differentiation and activation following viral infection. Notably, dendritic cells were also found to secrete IL1RN, targeting IFE cells, potentially mitigating the excessive immune damage

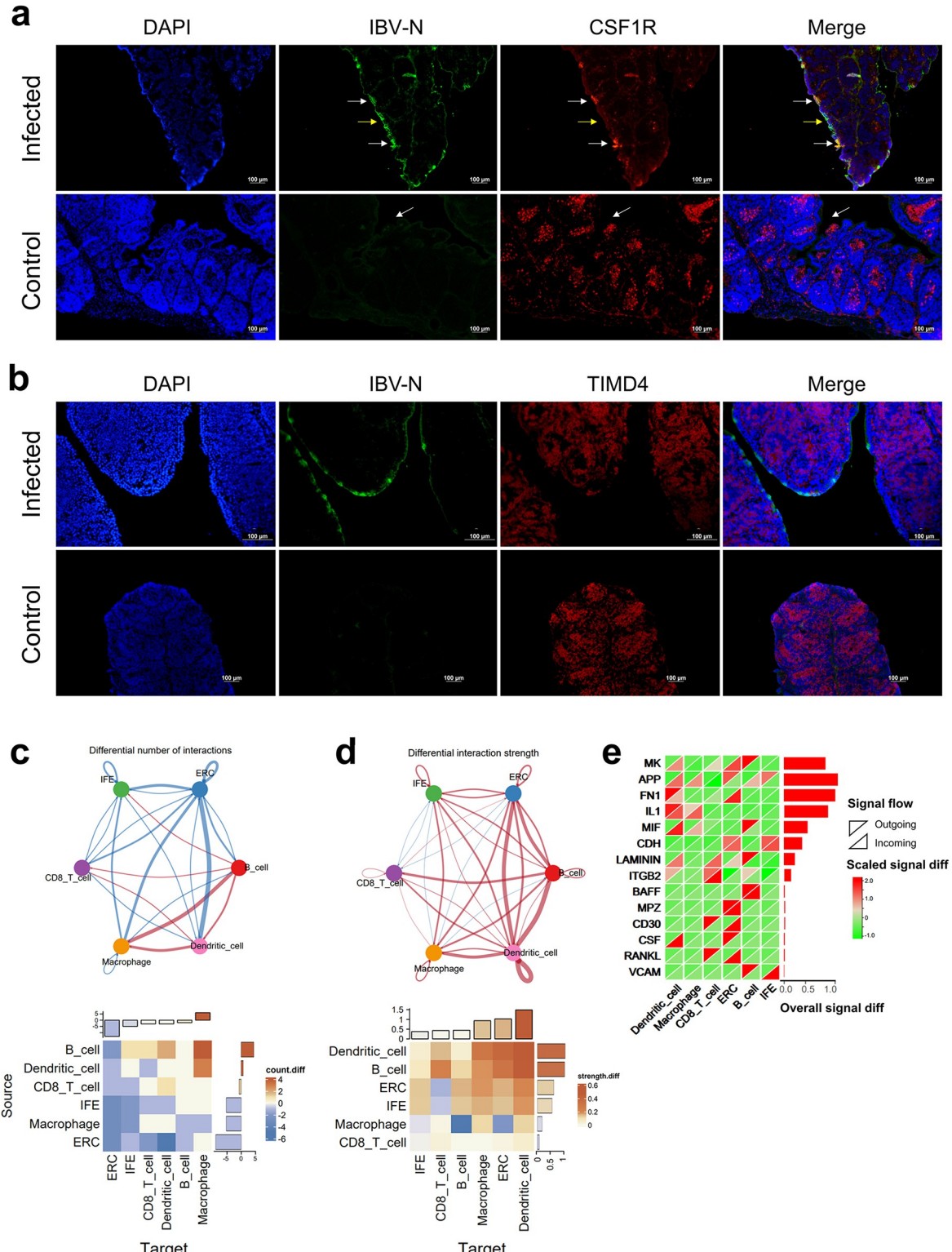

**Fig 5. Dual immunofluorescence identifying cell types infected by IBV in bursa and changes in cell-cell communication after IBV infection.** (a) Dual immunofluorescence identification of IBV infection in FAE cells; white arrows indicate the localization of IBV N protein in CSF1R-expressing FAE cells. Yellow arrows represent positive staining for IBV N protein in IFE cells. (b) Dule immunofluorescence detection of TIMD4 and IBV N protein. (c) Changes in cell communication numbers: The top network diagram shows cell clusters as nodes, with line thickness indicating changes in communication numbers. The lower heatmap details these changes,

with rows representing signal-sending cells and columns indicating signal-receiving cells. The color scale reflects the inter-group differences in signal communication frequency between different cell types (number of communications in the infected group—number in control group). The bar plots at the top and right side represent the overall differences in the number of signals sent/received by specific cell clusters. (d) Changes in communication strength: Similar to (c), with the top network diagram displaying changes in communication strength. (e) Inter-group differences in the communication strength of specific signaling pathways (receptor-ligand pairs) across cell clusters. Rows represent signal pathways and columns correspond to cell clusters, with heatmap colors depicting the strength variation of signals (infect group vs. control group). Upper left triangles for signal sent and lower right triangles for signal received. The bar plot on the right side shows the overall difference in communication strength of these signaling pathways between the IBV-infected group and the control group.

induced by IL1. IL21, secreted by CD8 T cells and acting upon themselves, plays a potential role in their activation. Moreover, CD8 T cells secrete IL22, influencing ERC cells, likely contributing to proliferation and repair post-injury. IFNγ, crucial in the antiviral response, is primarily secreted by CD8 T cells, affecting ERC, macrophages, dendritic cells, and CD8 T cells themselves (S7A Fig).

Subsequently, employing the MultiNicheNet algorithm, we constructed a ligand-target gene regulatory network among cell clusters. In IBV-infected bursa samples, we found dendritic cells as central regulators within this network, where signals sent by various cell types modulate the upregulation of IL6, IL1B, CXCL8, and matrix metallopeptidase 9 (MMP9) in dendritic cells. These genes play crucial roles in the activation, recruitment, and migration of immune cells. Dendritic cells also regulate immune responses by emitting signals such as IL15 and TGFB1. CD8 T cells contribute by releasing CCL5, CD28, and IFNγ, playing roles in immune cell recruitment and T cell activation processes (S7B Fig). Through two methods of cell communication analysis, we have obtained a comprehensive understanding of the intercellular signaling and regulatory interactions within bursa tissue during viral infection.

## Identification of cell clusters in the trachea tissues and cell population infected by IBV

To investigate the infection status of different cell types and immune responses in tracheal tissues after IBV infection, we collected tracheal tissues from chickens at 5 dpi from IBV-infected and control group. Single-cell RNA sequencing was performed using the 10x Genomics platform. After filtering out cells with high mitochondrial gene expression, multicellular clusters, erythrocytes, and chondrocytes expressing high levels of CYTL1, transcriptional profiles of 8992 cells were obtained (3162 infected group cells and 5830 control group cells). UMAP dimensionality reduction yielded 20 cell clusters (S2C Fig), which were identified based on marker gene expression to determine 10 distinct cell populations including FN1$^+$ fibroblasts (cluster 0,1,2,4,6,7,13,17), CYTL1$^+$ chondrocytes (cluster 11), KRT7$^+$ epithelial cells (cluster 9), S100B$^+$ ciliated cells (cluster 18), CSF1R$^+$ macrophages (cluster 8), CD3$^+$ T-cells (cluster 12), PECAM1$^+$ endothelial cells (cluster 5), RGS5$^+$ pericytes (cluster 14,16), MSC$^+$ muscle cells (cluster 3), and LOC395159$^+$ Schwann Cells (cluster 10) (Fig 6A and 6C). The proportion of T cells significantly increased after IBV infection indicating their recruitment (Fig 6B). Additionally, we observed the highest expression of viral 3'UTR RNA in epithelial cells (Fig 6C), indicating that IBV preferentially targets tracheal mucosal epithelial cells, which aligns with expectations.

## Changes of cell-cell communication in trachea after IBV infection

As mentioned above, to investigate the changes in intercellular interactions among different cell types, we first evaluated communications between cell populations through the CellChat algorithm, and ligand-receptor pairs with a P-value < 0.05 were included for analysis. We

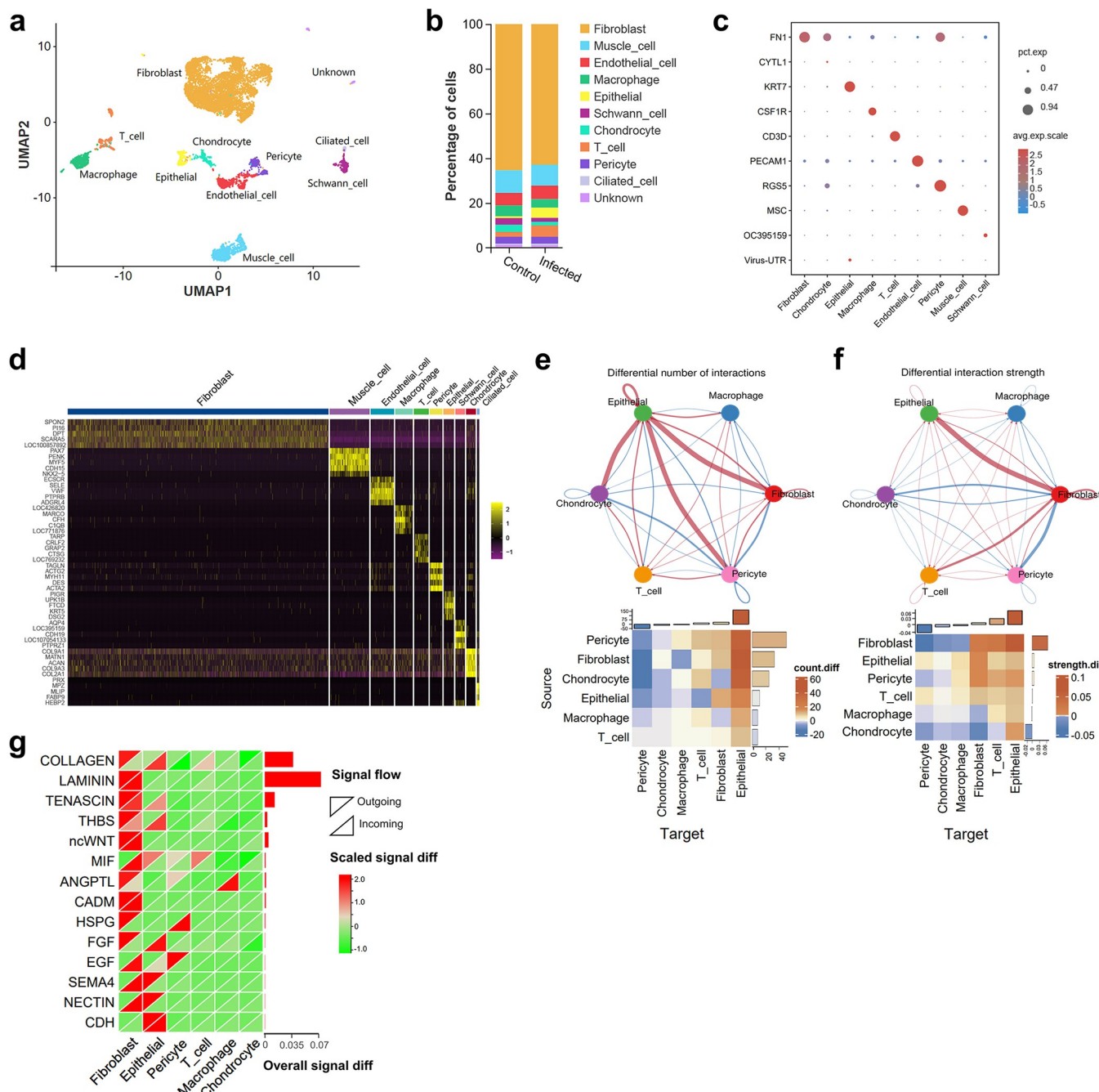

**Fig 6. Single-cell transcriptomic profiles of trachea tissues from IBV-infected and control chickens and changes of cell-cell communication after IBV infection.** (a) UMAP dimensional reduction and clustering, cells are categorized into subpopulations based on molecular specificity markers. (b) Bar graph showing the proportion of various cell types before and after infection. (c) Bubble plot illustrating the expression of cell type-specific molecular markers and viral RNA. The rows display gene names, and the columns represent cell populations. Bubble size indicates the proportion of gene expression in each cell type, while color denotes normalized gene expression levels(Z-score). (d) Heatmap depicting the expression levels (z-score) of characteristic genes in each cluster. (e) Changes in cell communication numbers: The top network diagram shows cell clusters as nodes, with line thickness indicating changes in communication numbers. The lower heatmap details these changes, with rows representing signal-sending cells and columns indicating signal-receiving cells. The color scale reflects the inter-group differences in signal communication frequency between different cell types (number of communications in the infected group—number in control group). The bar plots at the top and right side represent the overall differences in the number of signals sent/received by specific cell clusters. (f) Changes in communication strength: Similar to (e), with the top network diagram displaying changes in communication strength. (g) Inter-group differences in the communication strength of specific signaling pathways (receptor-ligand pairs) across cell clusters. Rows represent signal pathways and columns correspond to cell clusters, with heatmap colors depicting the strength variation of signals (infect group vs. control group). Upper left triangles for signal sent and lower right triangles for signal received. The bar plot on the right side shows the overall difference in communication strength of these signaling pathways between the IBV-infected group and the control group.

examined the changes in the number of cellular communications among cell populations after IBV infection. Our findings revealed a significant increase in signals sent from all cell populations, while only epithelial cells exhibited a significant increase in signals received (Fig 6E and S10 Table). Conversely, no significant changes of signal strength were observed between different cell populations (Fig 6F and S10 Table). Furthermore, there were no notable changes detected in signal strength within each pathway (Fig 6G and S11 and S12 Tables).

Further utilizing the MultiNicheNet algorithm, we initially identified the top 40 cellular communication interactions in the trachea sample. We observed that macrophages secrete IL1B, targeting fibroblasts and macrophages themselves. Additionally, through the ligands CCL1 and CCL4 acting on the CCR8 receptor of T cells, macrophages potentially play a role in the migration and chemotaxis of T cells. CD8 T cells can secrete IL21, CD6, CD28, and CCL1 signals, acting in an autocrine manner to promote self-activation. Notably, pericytes secrete CSF signals, acting on macrophages via CSF2R and CSF3R, potentially playing a role in the activation and recruitment of macrophages (S8A Fig). We then constructed a ligand-target gene regulatory network among cells. In T cells, autocrine secretion of IL21 promotes the upregulation of their own genes such as CCR8 and LAG3, regulating the immune response. Signals from epithelial cells, T cells, and macrophages, including HLA.A, CLDN3, and MPDZ, can activate the high expression of epithelial cadherin (CDH1) in epithelial cells, playing a role in maintaining epithelial barrier function and in the adhesion process of T cells [29]. Furthermore, T cell-derived IFNγ and macrophage-derived IL1B occupy central positions in the regulatory network, underscoring their critical roles in the antiviral immune process (S8B Fig).

## Weighted gene co-expression network analysis reveals immune response gene modules across tissues and cell populations

To further explore the immune response patterns across tissues and cell populations after IBV infection, as well as to identify key gene modules and hub genes, we integrated single-cell data from the kidney, bursa, and trachea tissues. Subsequently, a co-expression network was constructed using Weighted Gene Co-expression Network Analysis (WGCNA) to identify gene modules with conserved expression patterns across different tissues and cell populations. Initially, we further subdivided the already identified cell subpopulations within each tissue. Specifically, within kidney tissue, proximal tubular cells were divided into 9 subclusters while CD8 T cells were divided into 3 subclusters (S9A Fig). In bursa tissue, B cells were divided into 8 subclusters and IFEs were divided into 2 subclusters (S9B Fig). Similarly in tracheal tissue, fibroblasts were divided into 8 subclusters (S9C Fig). Next, we partitioned the gene expression data of all subpopulations across tissues into gene co-expression modules and constructed networks. A power value parameter of 6 was selected based on gene expression patterns (S10 Fig), followed by merging of modules with a correlation greater than 0.7. Ultimately, a total of 23 gene modules were identified (Fig 7A and 7B and S13 Table).

Functional enrichment analysis was performed on 23 gene modules, and 7 modules that play key roles in immune function were identified according to the enrichment results (Figs 7C and S11). These modules encompass diverse functions: the black module in T cell activation, Nod-like signaling, and differentiation; the dark grey module in MHC antigen presentation and cytokine signaling; the dark orange module in MHCI antigen presentation and Type I interferon response; the green module in B cell activation and BCR signaling; the green-yellow module in viral assembly and release; the lightcyan module in programmed cell necrosis and IL-6 response; and the steelblue module in Toll-like pathway, Type II interferon, and IL1 response. Notably, these modules exhibited a high correlation in expression patterns, especially

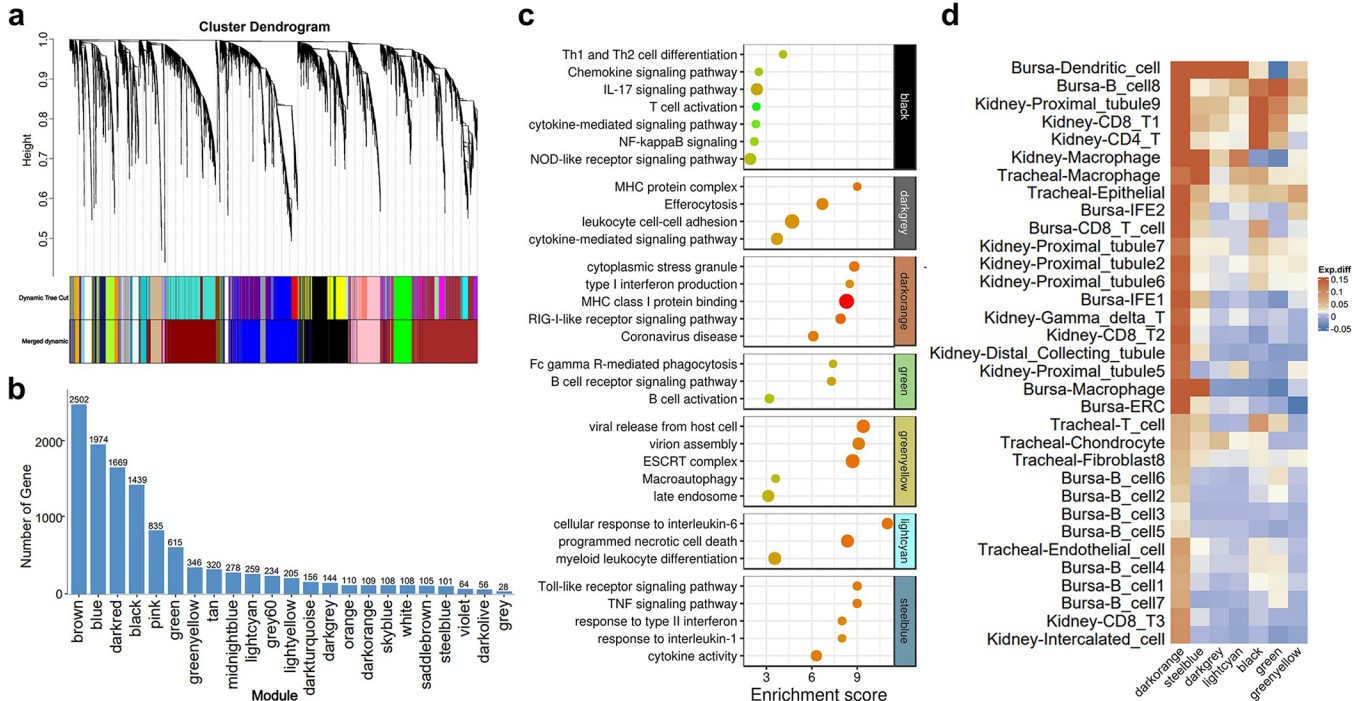

**Fig 7. Identification of gene modules and expression patterns via weighted gene co-expression network analysis (WGCNA).** (a) Module tree illustrating the hierarchical clustering of nodes (genes); vertical distances represent dissimilarities between nodes. Gene modules delineated through Dynamic Tree Cut are indicated, with modules merged based on a correlation threshold >0.7. (b) The number of genes in each module post-merging. (c) Enrichment results of immune-related gene modules. (d) Expression differences of immune-related gene modules across various cell clusters. Heatmap depicting the differences in gene module eigengene values between infected and control groups. The color scale represents the eigengene value differences for each gene module (infected group eigen- control group eigen). Columns representing gene modules and rows representing cell clusters. Numbers to the right of cell cluster names represent the subdivision of cell clusters.

between the lightcyan and steelblue modules, with a correlation coefficient of 0.7, indicating a consistent profile of immune function-related genes (S12 Fig).

Expression patterns of these modules were analyzed across various cell populations, based on module eigenvalues (S13 Fig and S14 Table). The steelblue, lightcyan, and darkgrey modules were highly expressed in macrophages and dendritic cells across tissues. Within these modules, the steelblue module was notably prevalent in dendritic cells and macrophages across all tissues. The lightcyan module was primarily expressed in bursal macrophages, while the darkgrey module was mainly expressed in tracheal macrophages. The greenyellow module primarily expressed in ERCs in the bursa, while the darkorange module was prevalent in all epithelial populations, including ERCs, IFEs, tracheal epithelial cells, renal proximal tubular cells, distal-collecting duct cells, and showed moderate levels in renal T cells. The black module was mainly expressed in T cells across tissues, and the green module was predominantly expressed in B-cells within the bursa (S13 Fig). The expression profiles of specific gene modules align with the known functions of the corresponding cell populations, thereby confirming the reliability of the data obtained.

We further evaluated the differential expression patterns of immune-related gene modules after IBV infection. The expression of the dark orange module notably increased across various cell populations, indicating its widespread role in antiviral functions. In macrophages and dendritic cells, the steelblue, darkorange, and lightcyan modules showed significant upregulation. The black and green modules demonstrated upregulation in T cells and renal proximal tubule cells, respectively, while the greenyellow module displayed increased levels in dendritic cells,

tracheal epithelial cells, among others (Fig 7D). The response of different cell populations in virus infection process vary; virus-infected cells can secrete interferons and chemokines to recruit immune cells, with macrophages primarily engaged in phagocytizing virus particles, T cells responsible for killing virus-infected cells and modulating immune responses, and B cells are responsible for producing antibodies, thereby activating humoral immunity. Our results systematically elucidate the cell population-specific expression patterns of immune gene modules and their variations following IBV infection, providing a comprehensive overview of the cellular response mechanisms.

## Identification and analysis of hub genes in immunologically relevant gene modules after IBV infection

Upon obtaining immunologically relevant gene modules, gene co-expression networks were constructed to identify hub genes within each gene module. Detailed descriptions are provided in the methods section and S14 Fig. Initially, we calculated the pairwise correlation of all genes within each module; summing the correlations between each gene and other genes within the module yielded the intramodular connectivity of each gene (S15 and S16 Tables). A higher connectivity of a gene indicates a higher correlation of its expression with that of other genes in the module, placing it in a core position within the co-expression network, known as a hub gene. To determine the differential expression of hub genes in IBV infected groups compared to control groups, we merged cell clusters exhibiting the largest differences in module eigengene expression (IBV infected groups vs. control groups), ranging from 10 to 12 clusters, and subsequently conducted differential expression analysis on these merged clusters, mapping the results to the nodes of the network (S17 Table). Expanding upon these foundations, we utilized the information from the Transcription Factors Database (TFDB) to identify and highlight the genes associated with transcription factors within the network (Fig 8). In the steelblue gene module, which is highly expressed in macrophages and dendritic cells, inflammatory factors such as IL1B and IL18, the avian defensin gene AVD, and transcription factors including CEBPB, MAFF, and PRDM1 are situated in central network positions and exhibit notably upregulated expression (Fig 8A). Within the darkorange module, which is commonly upregulated across various cell subgroups after IBV infection, transcription factors essential for antiviral processes such as IRF9 and STAT1 hold core positions and show significantly increased expression (Fig 8B). In the black gene module, highly expressed within T cells, transcription factors BCL11B, SATB1, TCF7, and TOX are central to the network, with pronounced upregulation (Fig 8C), playing a vital role in T cell activation and differentiation. We have also identified the hub genes within the lightcyan, green, greenyellow, and darkgrey modules (S15 Fig). In summary, the co-expression networks identified pivotal hub genes and transcription factors within immune gene modules, revealing their essential roles in orchestrating inflammatory responses, antiviral mechanisms, and immune cell activation.

## Differential gene expression and signaling pathway patterns of virus-infected cells and bystander cells in the IBV infected group

In the study conducted, we identified the target cell types of IBV: the distal-collecting duct cells in kidney, interfollicular epithelial cells (IFEs) and follicle-associated epithelial cells (FAEs) in bursa, and tracheal epithelial cells. To investigate gene expression differences and signaling pathway regulation in virus-infected cells at the single-cell sequencing level, we differentiated between virus-infected and non-infected cells within the IBV-targeted cell clusters. Due to the sequencing data not distinctly categorizing FAEs in the bursa samples as a separate cluster (Fig 4A), only the IFE cluster from the bursa was included in our analysis. Initially, we

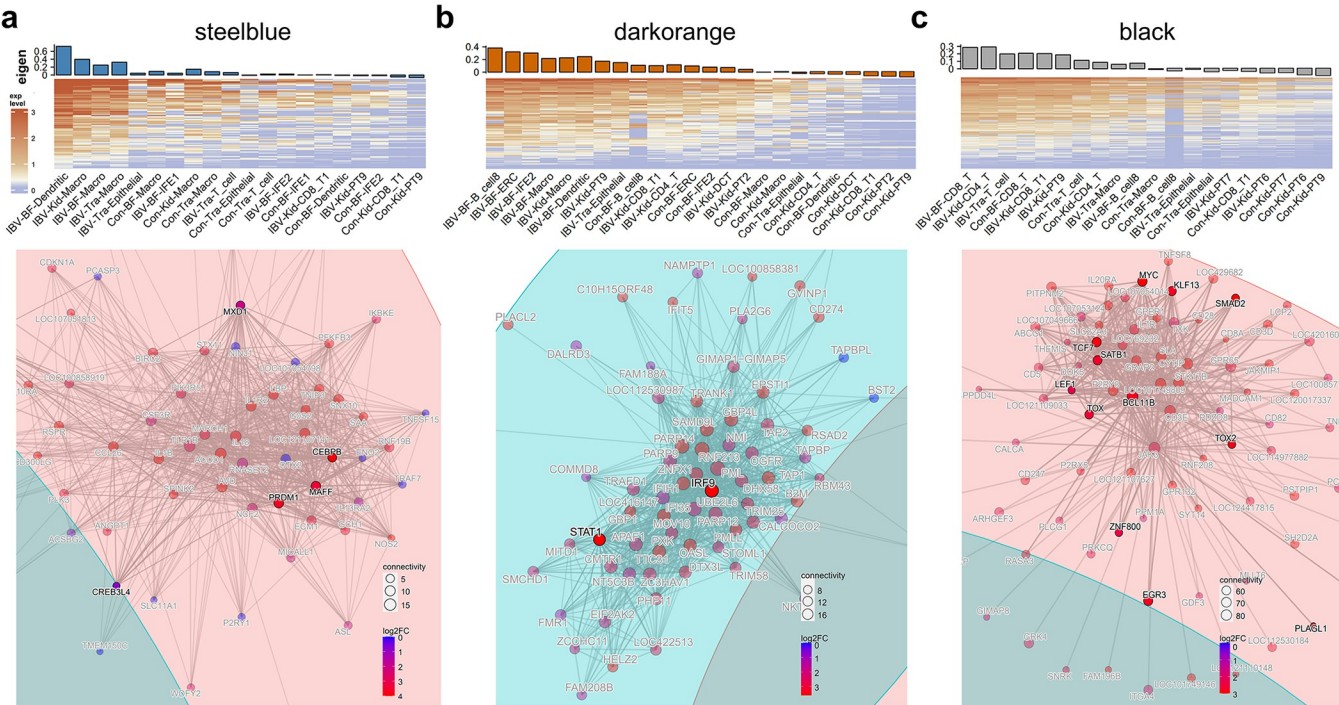

**Fig 8. Gene co-expression network.** (a) The top heatmap displays the expression patterns of genes within the steelblue gene module across different cell clusters in the virus-infected and control samples. Rows represent genes, and columns represent cell clusters. This heatmap displays the 10–12 cell clusters exhibiting the most significant upregulation of module eigen after viral infection. The bar plots above the heatmap represent the module eigen values for each cell cluster. Below, the Gene co-expression network diagram displaying nodes sized by gene connectivity within the module. Node colors represent the results of differential gene expression analysis after merging the cell clusters shown in the above heatmap (infected group vs. control group, p.adjust < 0.05).). $\log_2$FC = $\log_2$ (Infected group gene expression / Control group gene expression). Transcription factors are highlighted. Con, Control group; Kid, Kidney; IBV, IBV infected group; BF, Bursa of Fabricius; Tra, Trachea; PT, Proximal Tubule. Numbers to the right of cell cluster names represent the subdivision of cell clusters. (b) Gene expression heatmap and co-expression network of darkorange module. (c) Gene expression heatmap and co-expression network of genes in the steelblue module.

determined the viral load in the infected cells (Fig 9A and S1 Table). Subsequently, we assessed the differential gene expression patterns between virus-infected and non-infected cells of the same cell type within the IBV-infected group and conducted KEGG pathway enrichment analysis. Our findings revealed that in kidney tissue, chemokine CCL4, antiviral interferon regulatory factors IRF1, IRF8, IRF9, and interferon-induced genes IFIH6, IFIT5, IFIH1, among others, were upregulated in the infected distal-collecting duct cells compared to the non-infected 'bystander cells'. Moreover, genes crucial for water and salt reabsorption, such as AQP2 and SCNN1G, were downregulated in these cells (Fig 9B). In the IFE cells, infection was associated with the upregulation of interferon regulatory and inducible genes (IRF, IRF9, IFIT5, IFIH1), while a downregulation of IFI35 and IFI6 was observed, potentially indicating a mechanism to prevent excessive immune activation (Fig 9C). In infected tracheal epithelial cells, alongside interferon-induced and regulatory genes, upregulation of the interleukin receptors IL20A and the interleukin-binding protein IL18BP was noted (Fig 9D). Building on this, we identified a set of genes upregulated across infected distal-collecting duct cells, IFE cells, and tracheal epithelial cells. A total of 13 genes were found to be upregulated across all infected cell types (Fig 9E), including CD274, IFIH1, RSAD2, and IRF9, which were also identified as hub genes within the WGCNA gene co-expression network (Fig 8), underscoring their pivotal role in combating IBV infection. Enrichment analysis of upregulated genes in infected cells revealed the activation of pathways not only related to antiviral immunity but

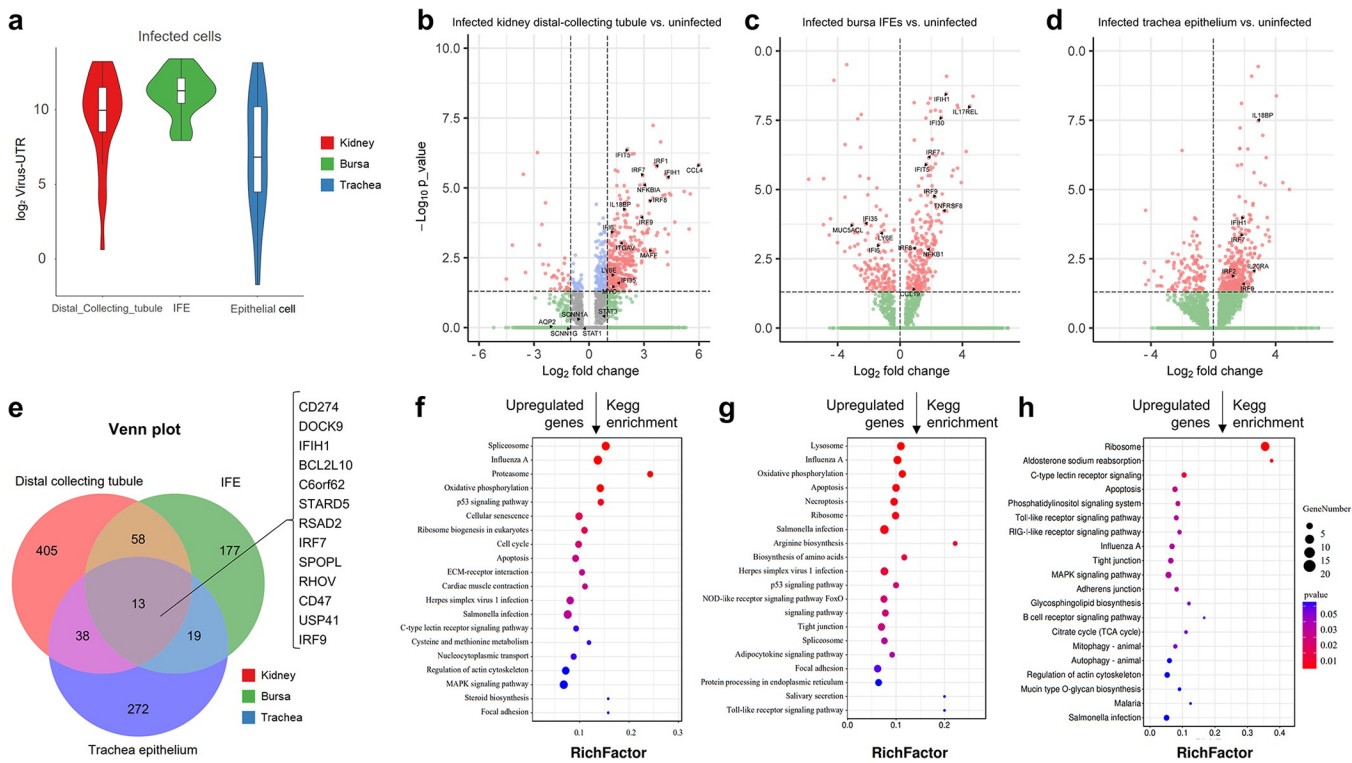

**Fig 9. Differential gene expression and pathway activation patterns in cells infected with IBV.** (a) Viral load in IBV-infected distal-collecting duct cells of kidney, IFEs of bursa, and tracheal epithelial cells. (b-d) Gene expression differences in IBV-infected distal-collecting duct cells, IFEs, and tracheal epithelial cells compared to their uninfected cells within the same cell clusters. (e) Venn diagram illustrating shared upregulated genes across infected distal-collecting duct cells, IFEs, and tracheal epithelial cells. (f-h) Enrichment results of KEGG pathways for genes upregulated in IBV-infected cells.

also those involving apoptosis, indicating a strong inducement of the apoptotic process by viral infection (Fig 9F, 9G and 9h).

## Discussion

Infectious bronchitis virus (IBV) exhibits broad tropism for mucosal epithelial cells, infecting the epithelial cells of the chicken respiratory tract, including nasal conchae, Harderian glands, and tracheal mucosa, causing severe respiratory symptoms. IBV also infects the kidneys, leading to severe nephritis, a primary cause of mortality. Many IBV strains establish infections in the gastrointestinal tract, including the duodenum, cecal tonsils, cloaca, and bursa [2]. Here, we employed single-cell RNA sequencing to comprehensively explore virus infection, inflammation and antiviral immune responses at the cellular level in multiple tissues post-IBV infection.

In this study, we performed 10x Genomics scRNA-seq on kidney, bursa, and tracheal tissues of IBV-infection and control groups to at 5 dpi explore inflammatory and antiviral immune responses in different tissues. Cell types were identified through marker gene expression patterns in various cell clusters. Consulting existing avian tissue single-cell sequencing literature [16,17,30,31], we observed significant physiological and gene expression differences between chickens and mammals (humans or mice), necessitating careful review and selection of cell markers for chicken tissue. We integrated histological features of chicken tissues, biological functions of molecular markers, and data on molecular markers identified in mammals to determine cell types, with most markers exhibiting high specificity. This is the first

identification of cell-type markers in chicken kidney tissue using single-cell sequencing data. Chicken kidneys predominantly comprise reptilian-type nephron, lacking loop of Henle, with direct connections between proximal and distal tubules [32], as evidenced by our data. LRP2 plays a crucial role in protein reabsorption in proximal tubule cells [33] and is highly expressed specifically in chicken kidney proximal tubules, as is cubilin (CUBN), another protein involved in reabsorption. Both genes serve as markers for chicken kidney proximal tubules. CALB1, a calcium-binding protein specific to distal tubules, and AQP2, a water channel protein specific to collecting ducts [34], previously used as markers for chicken kidney distal tubules and collecting ducts [35], also demonstrated good specificity in our data. For immune cell identification, we used CD4 and CD8 gene expression to determine CD4$^+$ and CD8$^+$ T-cells, respectively. Compared to mice and humans, chickens have a higher proportion of γδ T cells, up to 50% of peripheral blood lymphoid cells [36]. We identified a population of γδ T cells in the kidneys, using the previously identified marker gene KK34 [31,37]. Our data show that γδ T cells, after viral infection, exhibit a gene expression pattern distinct from CD4$^+$ and CD8$^+$ T-cells (S13 Fig), but with an upregulated gene pattern more akin to universally activated genes across all cell subgroups. Previews studies indicate that TLR stimuli like poly(l:C) can induce IFN-γ production in chicken γδ T cells [38]; however, our data suggest that IBV infection does not induce upregulation of IFN-γ in γδ T cells. The role of γδ T cells in viral infection resistance in chickens warrants further study.

Unlike mammals, the primary amino acid metabolites in avian species is urate rather than urea, with 80% of uric acid produced in the liver and excreted through the glomerular filtration and proximal tubular secretion in the kidneys [39]. Urate have low solubility and form crystals in the collecting duct due to urine concentration. The collecting duct secretes mucin to bind with uric acid salt crystals, maintaining a stable colloidal state in the urine, facilitating uric acid salt excretion [32]. Our data indicate significant cell death and a marked reduction in cell proportion in the collecting ducts following IBV infection, accompanied by a notable downregulation of mucin MUC4 gene expression, likely contributing to uric acid salt deposition post-IBV infection. Uric acid salts can also stimulate the kidneys, exacerbating inflammatory responses [40], leading to severe renal lesions and mortality in chickens.

The bursa, a unique immune organ in avian species, plays a crucial role in the maturation of B cells in humoral immunity. Our single-cell sequencing results of the bursa have identified various cell subpopulations. B cells, primarily expressing the avian B cell antigen Bu-1 (LOC396098), are the dominant cell type in the bursa. Previous studies have shown that bursa macrophages and dendritic cells specifically express TIMD4 and CSF1R, respectively [27]; these two marker genes also show high specificity in our data. ERCs in the bursal medulla form the structural framework, with our findings indicating their high expression of the chemokine CCL19, a marker for mammalian lymph node reticular fibroblasts, recruiting macrophages and T cells [28]. This suggests that bursal ERCs may play a role similar to reticular fibroblasts in the migration and homeostasis of immune cells in the avian bursal medulla [41]. Apart from serving as a primary immune organ, the bursa is also an important mucosal immune organ. The surface of the bursal folds is covered with a layer of mucin-secreting epithelial cells (IFE), while the FAE, covering the surface of the folded follicles, directly connects to the bursal medulla. In single-cell sequencing data, follicle-associated epithelium (FAE) cells were not clustered into a distinct cell population. This outcome is likely due to their gene expression patterns being more similar to those of reticular crypt epithelial cells (ERCs), resulting in their classification within the ERC cluster during dimensionality reduction and clustering process. Notably, both FAE and ERC originate from mesenchymal tissue. During developmental processes, ERCs in the follicles extrude from the epithelial basal membrane, with the topmost ERCs differentiating into FAEs [42,43]. Notably, within the ERC cluster,

several cells were identified expressing the FAE-specific marker gene CSF1R, underscoring the close relationship between these cell types (S5 Fig). FAE has antigen uptake capabilities; antigens or particles can enter the medulla through the FAE, and some pathogens can infect the FAE through its endocytic function to establish infection at the intestinal mucosa [27,44]. Our findings reveal that IBV can infect not only IFE cells but also target FAE cells, though its dependence on FAE endocytosis for infection requires further investigation. Our data, along with previous studies, suggest that certain IBV strains can damage the bursa, posing potential immunosuppressive risks [13,14]. In dual immunostaining for TIMD4 and IBV N protein, IBV N protein was not observed in macrophages expressing TIMD4. Furthermore, positive staining for IBV N protein was also absent in bursa follicles, which contains a high density of macrophages. Although viral RNA was indeed detected in macrophages in single-cell sequencing data, this could be attributed to the phagocytosis of viral particles or infected cells by macrophages, rather than direct infection. While some studies indicate that IBV can infect chicken macrophages, our data do not support this conclusion, potentially due to differences in the IBV strains used [45–47]. Considering many pathogens capable of causing immunosuppression can infect immune cells, further research is needed to confirm whether IBV possesses immunosuppressive capabilities. The absence of positive staining for IBV in the follicles of the bursa also suggests that IBV may not possess the capability to infect follicular T cells, B cells, and dendritic cells.

Since IBV primarily establishes infection in the respiratory tract, causing notable respiratory symptoms, it is essential to investigate the respiratory infection model and immune responses. In our sequencing data, chondrocytes and fibroblasts constituted the main cell groups. However, given that IBV predominantly infects tracheal mucosal epithelial cells, their role in mucosal antiviral immunity is relatively limited. Therefore, enriching cells from the tracheal mucosal surface is necessary. Indeed, virus-infected cells are fragile; during the preparation of single-cell sequencing samples, these cells are prone to death and often filtered out during data quality control, necessitating careful optimization of single-cell library preparation methods.

Based on the single-cell expression profiles obtained, we utilized the R package CellChat [23] and Multinichenet [24] to study the interactions between different cell types in the infected and control groups, delineating a global map of cell communication. We observed a significant upregulation of the SPP1 pathway, also known as Osteopontin (OPN), following renal infection. SPP1, a key mediator in inflammatory responses, primarily acts as a chemotactic factor for immune cells and induces cytokine secretion. On the one hand, SPP1 can promote T cells to secrete IFN-γ [48,49], and on the other, it acts as an inhibitor to prevent excessive T cell activation [50,51]. In our data, the SPP1 signal was predominantly emitted by renal tubular cells and received by T cells, indicating an immunomodulatory role of renal tubular cells on T cells post-IBV infection.

Macrophage migration inhibitory factor (MIF), a pro-inflammatory cytokine, can be secreted by various cell types, causing macrophage retention at inflammation sites and promoting the production of pro-inflammatory factors like IL1B [52]. Extracellular MIF binds to C-X-C chemokine receptors and activates chemokine responses in synergy with CD74, while intracellular MIF participates in toll-like receptor and NLRP3 inflammasome-mediated inflammatory responses [53,54]. Previews studies show that post-SARS-COV-2 infection, MIF expression is significantly upregulated, and serum MIF levels serve as an important prognostic marker for SARS-COV-2 infection [55–57], with MIF also acting as a stimulatory molecule in renal inflammation [58]. In our data, communication strength of MIF in renal and bursa tissues were significantly upregulated, with signals primarily sent by renal tubular cells. Interestingly, in the bursa, MIF was mainly secreted by B cells, which are also capable of secreting MIF

in humans, though its biological function remains underexplored [59]. In the bursa, dendritic cells also served as receivers of the MIF signal. The amyloid precursor protein (APP) pathway was significantly upregulated in the kidneys and bursa following viral infection. In the kidneys, amyloid proteins can damage the filtration system, leading to kidney failure [60]. APP also plays a significant role in anti-infective immunity, inhibiting the replication of HSV-1 and Influenza A virus and stimulating macrophages to secrete pro-inflammatory cytokines [61,62]. In our data, APP was mainly secreted by renal tubular cells and bursa epithelial cells, and received by macrophages and dendritic cells, suggesting its role in combating IBV infection. Additionally, post-SARS-Cov-2 infection, the APP pathway is also upregulated, potentially exacerbating neuroinflammation [63]. Furthermore, we performed cell communication analysis using the Multinichenet algorithm, which, compared to CellChat, integrates the activation of ligand-receptor pairs and downstream target genes to provide a comprehensive landscape of signaling cascade activation. Our results reveal the pivotal roles of macrophages, dendritic cells, and CD8 T cells emerge as central regulators, modulating immune cell recruitment, activation, and antiviral effector mechanisms through the strategic secretion of cytokines (IL1B, IFNγ, IL6, IL15), chemokines (CCL1, CCL5), and growth factors (CSF1, CSF3). The avian CD8 T cell subset exhibits unique characteristics, acting as a prominent source of IL-21 and IFNγ, potentially reflecting evolutionary adaptations within the avian immune system. Furthermore, the regulatory networks highlight the intricate cross-talk between immune and structural cells, underscoring the importance of epithelial barrier integrity and tissue repair processes mediated by signals such as CDH1 and IL-22. Overall, our data reveal the intercellular interaction map post-IBV infection, providing a basis for selecting antiviral targets.

Weighted gene co-expression network analysis (WGCNA) is a systems biology method used to describe patterns of correlative gene expression in transcriptome samples. WGCNA identifies highly correlated gene modules, finds hub genes through connectivity, and uncovers gene expression pattern correlations. It has widespread applications in identifying candidate biomarkers, therapeutic targets, and genes critical in biological processes [64]. WGCNA is also extensively employed in infectious disease research [65,66]. After obtaining single-cell global transcriptomic profiles post-IBV infection from multiple tissues, we integrated these data to identify homogeneity and heterogeneity in immune responses induced in different tissues and cell types. Our WGCNA analysis revealed seven immune response-related modules, categorized into five types: modules universally upregulated post-infection (darkorange); gene modules expressed mainly in macrophages and upregulated post-infection (darkgrey, lightcyan, steelblue); modules primarily expressed in epithelial cells (greenyellow); modules highly expressed in T cells (black); and those highly expressed in B cells (green). Furthermore, we constructed a gene co-expression network, mapping gene expression changes post-viral infection and transcription factor information from the TFDB [67] onto the network. This approach unveiled several key antiviral immune genes, including some unique to avian species and unnamed. In the steelblue module, classical pro-inflammatory factors IL1B and IL18 [68] were central and upregulated. Additionally, avidin (AVD), unique to avian species and upregulated, was central in the network. AVD, known for its broad-spectrum antibacterial activity, is strongly stimulated by interferon α, but its role in viral infections remains unclear [69,70]. Interestingly, we found the avian-specific non-coding RNA LOC121107141 at the network's core and upregulated, suggesting its potential function in antiviral immunity. The vertebrate interferon regulatory factor (IRF) family, comprising 11 members, plays various roles in multiple biological processes, including antiviral defense, and regulation of cell proliferation, differentiation, and apoptosis. In the universally upregulated gene module (darkorange) post-infection, the transcription factor IRF9 was central. Notably, chickens lack the mammalian homolog of IRF9, the annotated chicken IRF9 is actually homologous to IRF10 [70], absent in

primates and rodents [71]. Mammalian TLR signaling cascades downstream components include IRAK-1, IRAK-4, TRAF-6, NF-κB, and IRF family members. However, chicken TLR4 activation does not effectively induce type II IFN responses, and chickens lack TRAM in their genomic resources [72,73], making it necessary to thoroughly study chicken IRF9 to confirm its function. Additionally, STAT1, central in the network, has been shown in prior research to be downregulated ectopically by IBV to delay IFN response [74], potentially a mechanism for IBV to counteract the host immune system. In the black module highly expressed in T cells, several transcription factors were central and upregulated, including BCL11B, SATB1, LEF1, TOX, TOX2, TCF7, MYC, etc. Studies in mammals have shown these genes play significant roles in T cell development, differentiation, lineage determination, and activation [75–80]. Avian leukosis virus (ALV) infection has been shown to significantly reduce BCLL11B expression in chicken spleens [81]. In MDV-susceptible birds, TCF7 expression is significantly reduced [15]. The roles of these genes in antiviral immunity remain to be investigated.

In summary, through scRNA-seq analysis, we have revealed key factors in the pathogenesis of IBV infection in chickens, verified the target cells of IBV infection, and provided data support for further elucidation of IBV pathogenicity. More importantly, our analysis identified several pathways and key genes potentially functioning in the avian immune system. While some pathways and molecules have been characterized in mammals, others are unique to avian species and largely unstudied. We hope our scRNA-seq data will also bring new insights into avian immunological research.

## Materials and methods

### Animals and ethics statement

Specific pathogen free (SPF) chickens were purchased from Beijing Boehringer Ingelheim Vital Biotechnology Co., Ltd. (Beijing, China). The Beijing Administration Committee of Laboratory Animals approved the animal experimental protocols under the auspices of the Beijing Association for Science and Technology (approval SYXK [Jing] 2018–0038) and Ethical Censor Committee at China Agricultural University (approval 2022188).

### Virus and experimental animal infection

The SD subtype of the Infectious Bronchitis Virus (IBV) was isolated and identified by our team [82]. We assigned eight one-day-old, specific pathogen-free (SPF) chickens randomly into two groups: the IBV-infected group and the control group, with four chickens in each. The IBV-infected group received an inoculation of 100 μL containing $10^5$ EID$_{50}$ of IBV, while the control group received PBS as a negative control, both administered via an eye dropper. Five days post-inoculation, we dissected the chickens from both groups. Tissue samples were collected for single-cell RNA sequencing, histopathological staining, and immunofluorescence analysis. One-day-old SPF chickens were selected for inoculation and sampled at 5 days post-inoculation (dpi) because IBV initially replicates in the trachea before spreading to the kidneys and bursa. At this time point (5 dpi), tissues from the trachea, kidneys, and bursa are concurrently infected by the virus, enabling the sampling and single-cell sequencing analysis of various tissues from the same infected chicken.

### Histopathological staining

Kidney, bursa, and tracheal tissues from both infected and control groups were fixed in 4% paraformaldehyde for 72 hours. Following fixation, the tissues underwent dehydration in ascending grades of ethyl alcohol, clearing in xylene, and embedding in paraffin wax. Tissue

**Table 1. Antbodies and reagents used in present study.**

| Reagent | Dilution | Source |
|---|---|---|
| Rabbit pab AQP2 | 1:100 | Abmart (PK58037) |
| mouse mab calb1 | 1:200 | Boster Bio (BM0203) |
| Mouse mab CSF1R | 1:100 | Bio-rad (MCA5956) |
| Mouse mab Tim4 | 1:500 | Bio-rad (MCA6407) |
| Mouse mab IBV-N | 1:500 | HyTest (3BN1) |
| goat anti mouse 488 | 1:500 | Thermofisher (A48286) |
| goat anti rabbit 555 | 1:500 | Cell Signaling Technology (4413S) |
| Block buffer | - | Beyotime Biotechnology (P0231) |
| Primary antibody dilution buffer | - | Beyotime Biotechnology (P0276) |
| mIHC kit (dual immunofluorescence) | - | Panovue |
| PAP Pen | - | Absin (abs929) |

cross-sections of 4 μm thickness were cut using a LEICA RM 2015 microtome, mounted on slides, deparaffinized in xylene, and rehydrated through descending grades of ethyl alcohol. The sections were then stained with hematoxylin-eosin, mounted with neutral balsam, and examined using an Olympus BX51 & DP 7 Digital Camera System.

## Immunofluorescence

**Dual immunofluorescence for bursa of Fabricius.** Fresh tissue samples were cryosectioned at 8μm onto Poly-L-Lysine slides after embedding in OCT compound. The sections were fixed in 100% methanol for 10 minutes at 4°C and air-dried for 1 hour at room temperature. After blocking for 1 hour, primary antibodies (listed in Table 1) were applied and incubated overnight at 4°C. Sections were then washed three times with TBST, followed by incubation with secondary antibodies for 20 minutes, TSA enhancing solution for 10 minutes, and washing according to the manufacturer's instructions. An antibody stripping step for 15 minutes and TBST wash preceded the second round of primary antibody staining, following the same protocol as the first round. After the second staining round, slides were coverslipped, and fluorescence signals were captured using a Nikon A1 confocal microscope.

**Dual immunofluorescence for kidney.** Tissues were embedded in paraffin wax and cut into 4 μm thick cross-sections. The sections were then mounted on slides, deparaffinized in xylene, and rehydrated through descending grades of ethyl alcohol. Heat-induced antigen retrieval was performed using citrate-EDTA solution (Beyotime) for 10 minutes. The process followed for blocking, antibody incubation, washing, and subsequent steps was identical to that described for the bursa.

**Single immunofluorescence.** Tissues embedded in paraffin wax were sectioned into 4 μm thick cross-sections. After rehydration, heat-induced antigen retrieval was performed, followed by overnight incubation with primary antibodies. The sections were then incubated with fluorescent secondary antibodies and subsequently coverslipped for observation.

## Sample and library preparation for 10x scRNAseq

Cellular suspensions were processed using the 10X Genomics Chromium Next GEM Single Cell 3' Reagent Kits v3.1 to generate single-cell Gel Bead-In-Emulsions (GEMs). Upon Gel Bead dissolution in a GEM, primers containing Illumina R1 sequence, a 16 nt 10x Barcode, a 10 nt Unique Molecular Identifier (UMI), and a poly-dT primer sequence were released and mixed with cell lysate and Master Mix. Full-length, barcoded cDNAs were reverse-transcribed

from poly-adenylated mRNA. Silane magnetic beads were used to remove residual reagents and primers from the post-GEM reaction mixture. The barcoded cDNAs were amplified via PCR for library construction. The R1 sequence was added during GEM incubation, while P5, P7, sample index, and R2 sequence were incorporated during End Repair, A-tailing, Adaptor Ligation, and PCR. The final libraries, containing P5 and P7 primers, were suitable for Illumina bridge amplification. The resulting Illumina-ready sequencing libraries comprised standard paired-end constructs, with the 10x Barcode and UMI encoded in Read 1, cDNA fragment in Read 2, and sample index in the i7 index read.

## 10x scRNAseq sequencing data analysis

Raw BCL files were converted to FASTQ files, aligned, and quantified for counts using 10X Genomics Cell Ranger software (version 3.1.0). Cell-by-gene matrices for each sample were imported into Seurat version 4.0 for downstream analysis [83]. Cells exhibiting either an unusually high number of UMIs ($\geq$8000) or a mitochondrial gene percentage ($\geq$10%) were excluded. Additionally, cells with fewer than 500 or more than 4000 genes detected were also omitted. Doublet GEMs were identified and filtered out using DoubletFinder (v2.0.3), which involved generating artificial doublets and assessing each cell's proportion of artificial k nearest neighbors (pANN), ranked according to the anticipated number of doublets. To minimize batch effects and condition-specific biases on clustering, Harmony was utilized. This algorithm projects cells into a shared embedding, enabling them to group by cell type rather than dataset-specific conditions, to aggregate samples [84]. For cluster visualization, uniform manifold approximation and projection (UMAP) was generated using the same principal components [85]. For quantification of viral RNA in cells, we used SoupX to remove contaminating background viral RNA from single-cell droplets [86].

## Cell-cell communication analysis

**CellChat.** The single-cell gene expression matrix and cell group information were inputted to simulate the probability of intercellular communication. This simulation integrated gene expression with interactions between signaling ligands, receptors, and their cofactors, as delineated in the CellChat software [23]. CellChatDB, utilized in this analysis, incorporates signaling molecule interaction information sourced from the KEGG Pathway database and recent experimental studies. The communication probability for each signaling pathway was calculated in relation to the sum of the ligand-receptor pairs. This calculation facilitated the determination of the contribution degree. Based on this contribution degree, key ligand-receptor pairs instrumental in each pathway were identified. Furthermore, horizontal communication differences among receptor pairs across various groups and cell populations were compared using the software. In the standard output of CellChat for cell communication analysis, the communication probability (p) for all ligand/receptor pairs (i.e., signaling pathways) is first calculated. A pathway is considered to be actively communicating if p-value $<$ 0.05. This allows for the determination of the total number of signaling communications occurring within each cell subpopulation. Based on this, the signaling output strength and signaling reception strength for specific signaling pathways in each cell cluster can be obtained. For a given signaling pathway, the sum of signaling output strengths across all cell groups is equal to the sum of signaling reception strengths, serving as the pathway's overall activation strength in the sample. For a specific cell subpopulation, the sum of signaling output strengths across all pathways constitutes the overall signaling output strength of that subpopulation, and similarly for signaling reception strength. The CellChat algorithm can perform statistical tests on cell communication across different groups to identify pathways with statistically significant

differences. By subtracting the signal intensities of different groups, one can determine the difference in signaling strength between specific cell groups or specific signaling pathways across different groups (in our data, this would be the IBV infected group—control group). The data were visualized by heatmap using the R packages ComplexHeatmap [87] and TBtools [88].

**Multinichenet.** This algorithm incorporates the expression of ligand genes in signaling sender cells, receptor genes in signaling receiver cells, and downstream target genes based on the nichenetv2 network developed by the authors, uses a weighted approach to calculate communication probabilities between cell populations [24]. It notably facilitates comparisons across different conditions, such as between infected and control groups in our study. We evaluated the differential cell communication patterns in kidney, bursa, and trachea tissues between IBV-infected and uninfected groups. Potential ligands were extracted if they were expressed in at least 5% of the sending cells within their respective clusters, with a p-value cutoff set to 0.05. We prioritized predicted ligand-receptor interactions using the default prioritization weights and visualized the top 40 interactions in both IBV-infected and uninfected groups using circus plots (S6S Fig). Subsequently, intercellular signaling cascade communication networks were constructed based on the expression correlation of ligand-target genes and their differential expression patterns between IBV-infected and control groups (S6B Fig). Specifically, the differential expression threshold was set to a $\log_2$ fold change (FC) $> 0.5$, a p-value $< 0.05$, and a cell expression proportion $> 0.05$. For visualization purposes, in kidney tissue, the top 100 genes with regulatory capacity in the ligand-target gene regulatory network were included, with either Spearman or Pearson correlation coefficients greater than 0.6. In bursa tissue, the top 50 genes with regulatory capacity were included, with both Spearman and Pearson correlation coefficients greater than 0.85. For trachea tissue, the top 250 genes with regulatory capacity were included, with either Spearman or Pearson correlation coefficients greater than 0.5.

## Enrichment analysis

Enrichment analysis was conducted using the enrichGO and enrichKEGG functions of the R package clusterprofiler. This analysis focused on genes upregulated or downregulated by at least two-fold in the IBV-infected group compared to the control group, with a significance threshold of q $< 0.05$. GO and KEGG enrichment results were merged and visualized as network graphs using the emaplot function, adhering to the selection criterion of q $< 0.05$. Following the identification of gene modules through WGCNA, all genes within these modules underwent enrichment analysis and network visualization.

## Weighted gene co-expression network analysis (WGCNA)

We have drawn a schematic diagram of the WGCNA process for ease of understanding (S14 Fig). Specifically, cell clusters from kidney, bursa, and tracheal tissues were further subdivided. Each cell cluster was further categorized into IBV-infected and control groups, excluding unidentified cell clusters. Consequently, a total of 66 cell clusters were obtained (33 from the IBV-infected group and 33 from the control group) (S9 Fig). Co-expression networks were constructed utilizing the WGCNA R package (v1.47) [89]. Following gene filtration, gene expression levels were imported into WGCNA for co-expression module construction. This was achieved using the blockwiseModules function with default settings, except for modifications where the power was set to 8, TOM Type to unsigned, merge-Cut-Height to 0.7, and minModuleSize to 50. This process clustered genes into 23 modules. Intramodular connectivity (K.in) and module correlation degree (MM) for each gene were calculated. Genes exhibiting high connectivity were identified as potential hub genes, likely possessing significant

functions. Network visualization was accomplished using the emaplot function of the Cluster-Profiler package [90]. Following the identification of gene modules, the functions of 23 distinct gene modules were elucidated via enrichment analysis. Of these gene modules, 18 exhibited significant enrichment of functional categories, with seven modules related to immune response and inflammatory processes according to the enrichment analysis results. The differential gene expression patterns of the immune-related modules in the infection group were derived by subtracting the control group values from those of the infection group (IBV group gene module eigen–Control group gene module eigen). Building upon this, the top 10–12 cell clusters exhibiting upregulation within the immune-related gene modules after infection were merged, yielding seven integrated cell clusters. Differential gene expression analysis was conducted on these integrated clusters (IBV group vs. Control group), and the results were visualized by mapping the expression changes onto node colors. Transcription factor genes, as obtained from the TFDB database [67], were highlighted to facilitate the identification of regulatory elements within these clusters.

## Differentially expressed genes analysis

The expression value of each gene within a given cluster was compared against the rest of the cells using the Wilcoxon rank sum test. Identification of significantly upregulated genes was based on several criteria: Fold change $>2$, cell expressing percentage $>0.1$, q-value $<0.05$.

## Supporting information

**S1 Fig. Immunofluorescence in chicken tissues infected with IBV.** Detection using an antibody against IBV N protein, with virus-positive areas indicated by green fluorescence. HG, Harderian gland; CT, Cecal tonsil.
(PDF)

**S2 Fig. UMAP dimensionality reduction and cell clustering.** (a) Kidney tissue cell clusters, resolution = 0.85. (b) Bursa tissue cell clusters, resolution = 0.6. (c) Trachea tissue cell clusters, resolution = 0.5.
(PDF)

**S3 Fig. Functional enrichment of differential genes in kidney distal-collecting tubule cells (IBV vs. Control).** (a) Enrichment of upregulated genes after infection. (b) Enrichment of downregulated genes after infection. $\log_2 FC>2$, q-value$<0.05$.
(PDF)

**S4 Fig. Volcano plot of differential gene expression analysis.** The figure shows the results of differential gene expression analysis (infection group vs. control group) in kidney tissues of (a) distal convoluted tubule cells, (b) CD8 T cells, and (c) macrophage clusters, comparing the infected group to the control group.
(PDF)

**S5 Fig. Distribution of CSF1R-expressing cells (black) of bursa tissue.**
(PDF)

**S6 Fig. MultiNicheNet identified differential cell-cell communication patterns between kidney tissues from infected and uninfected chickens.** (a) The circus plot illustrates the top 40 differential ligand-receptor pairs between IBV-infected and uninfected chicken kidneys, categorized by cell type. The directionality from sender to receiver cell types is indicated by arrows, with the color of each arrow denoting the cell type expressing the ligand. (b) The intergroup differential intercellular signaling communication network predicted by

MultiNicheNet. This network displays the potential divergent cell-to-cell signaling cascade patterns between IBV-infected and uninfected groups. This network is composed of predicted ligand-target associations, where "upregulated" links signify a positive correlation in expression between ligand-receptor pairs and target genes in receptor cell types. Conversely, "downregulated" links indicate a negative correlation. Target genes, identified as the top 100 genes regulated by specific ligands with the highest regulatory potential within the nichenet network, exhibit expression correlation with specific upstream ligand-receptor pairs (Pearson or Spearman correlation $> 0.6$).
(PDF)

**S7 Fig. MultiNicheNet identified differential cell-cell communication patterns between bursa tissues from infected and uninfected chickens.** Similar to S2a Fig. (a) The circus plot illustrates the top 40 differential ligand-receptor pairs between IBV-infected and uninfected chicken bursa, categorized by cell type. (b) The inter-group differential intercellular signaling communication network predicted by MultiNicheNet. Target genes, identified as the top 50 genes regulated by specific ligands with the highest regulatory potential within the nichenet network, exhibit expression correlation with upstream ligand-receptor pairs (Pearson and Spearman correlation $> 0.85$).
(PDF)

**S8 Fig. MultiNicheNet identified differential cell-cell communication patterns between trachea tissues from infected and uninfected chickens.** Similar to S2a Fig. (a) The circus plot illustrates the top 40 differential ligand-receptor pairs between IBV-infected and uninfected chicken trachea, categorized by cell type. (b) The inter-group differential intercellular signaling communication network predicted by MultiNicheNet. Target genes, identified as the top 250 genes regulated by specific ligands with the highest regulatory potential within the nichenet network, exhibit expression correlation with upstream ligand-receptor pairs (Pearson or Spearman correlation $> 0.5$).
(PDF)

**S9 Fig. Cell subcluster for weighted gene co-expression network analysis.** (a) In kidney, proximal tubule cells are subdivided into 9 distinct subclusters, CD8 T cells into 3 subgroups. (b) In bursa, B cells are divided into 8 subclusters. (c) In tracheal, fibroblasts are divided into 8 subclusters.
(PDF)

**S10 Fig. Scale-free topology model fit (R2) and soft-thresholding power of WGCNA.** Weighted gene co-expression network showing the relationship between scale-free topology model fit (R2) and soft-thresholding power, depicted on the left; and the relationship between mean connectivity and soft-thresholding power, depicted on the right.
(PDF)

**S11 Fig. Functional enrichment analysis of genes in modules identified by WGCNA (q-value $< 0.05$).**
(PDF)

**S12 Fig. Heatmap of inter-module correlations.** The heatmap shows the correlation coefficient among gene modules.
(PDF)

**S13 Fig. Expression patterns of gene modules (eigen value) in different cell populations.**
(PDF)

**S14 Fig. WGCNA analysis flow chart.**
(PDF)

**S15 Fig. WGCNA gene co-expression network of lightcyan, green and greenyellow module.** Similar to Fig 8. Network diagram displays nodes sized by gene connectivity within the module, with node color indicating the results of differential gene expression analysis(p. adjust<0.05). $\log_2FC = \log_2(IBV/Control)$. Transcription factors are highlighted.
(PDF)

**S1 Table. Cell Types infected by IBV in various tissues.** The table present the quantity, proportion, and average viral RNA load of IBV-infected cells within distinct cell clusters in kidney, bursa, and tracheal tissues, respectively.
(XLSX)

**S2 Table. Differential gene expression analysis of kidney distal-collecting tubule cells, CD8 T-cells and macrophages (IBV VS. Control).**
(XLSX)

**S3 Table. Cell communication numbers and probabilities between cell populations in kidney.**
(XLS)

**S4 Table. Probabilities (strength) of signal emission and reception in specific signaling pathways between cell populations in kidney, including differences in communication probabilities between Infected and control groups.**
(XLS)

**S5 Table. Probabilities of receptor-ligand pair communications in specific cell communication pathways in kidney.**
(XLS)

**S6 Table. Prioritization of signaling pathways in cell communication analysis using multinichenet.** The table show the prioritization of the signaling pathways in the kidney, bursa, and trachea tissues from the infected and control groups, respectively.
(XLSX)

**S7 Table. Cell communication numbers and probabilities between cell populations in bursa.**
(XLS)

**S8 Table. Probabilities (strength) of signal emission and reception in specific signaling pathways between cell populations in bursa, including differences between Infected and control groups.**
(XLS)

**S9 Table. Probabilities of receptor-ligand pair communications in specific cell communication pathways in bursa.**
(XLS)

**S10 Table. Cell communication numbers and probabilities between cell populations in trachea.**
(XLS)

**S11 Table. Probabilities (strength) of signal emission and reception in specific signaling pathways between cell populations in trachea, including differences in communication**

probabilities between Infected and control groups.
(XLS)

**S12 Table. Probabilities of receptor-ligand pair communications in specific cell communication pathways in trachea.**
(XLS)

**S13 Table. Expression levels of all genes within modules across different cell populations.**
(XLS)

**S14 Table. Expression patterns (eigen) of gene modules in different cell populations.**
(XLS)

**S15 Table. Expression correlations(weights) among all genes within immune-related gene modules.**
(XLSX)

**S16 Table. Connectivity of all genes within immune-related gene modules.**
(XLS)

**S17 Table. Differential gene expression analysis of genes within immune-related gene modules (IBV vs. Control).**
(XLSX)

## Author Contributions

**Conceptualization:** Jing Zhao, Guozhong Zhang.

**Formal analysis:** Chengyin Liukang, Jing Zhao, Min Huang, Rong Liang, Ye Zhao, Guozhong Zhang.

**Funding acquisition:** Guozhong Zhang.

**Investigation:** Chengyin Liukang.

**Methodology:** Chengyin Liukang, Jing Zhao, Jiaxin Tian, Min Huang, Rong Liang.

**Project administration:** Ye Zhao.

**Resources:** Ye Zhao, Guozhong Zhang.

**Software:** Chengyin Liukang, Jiaxin Tian, Rong Liang.

**Supervision:** Guozhong Zhang.

**Validation:** Ye Zhao, Guozhong Zhang.

**Visualization:** Chengyin Liukang, Jing Zhao.

**Writing – original draft:** Chengyin Liukang, Jing Zhao, Jiaxin Tian, Min Huang.

**Writing – review & editing:** Jing Zhao, Guozhong Zhang.

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
