## [Decision Letter · Decision Letter 0]

2 Feb 2024

Dear Prof. Zhang,

Thank you very much for submitting your manuscript "Deciphering Infected Cell Types, Hub Gene Networks and Cell-Cell Communication in Infectious Bronchitis Virus via Single-Cell RNA Sequencing" for consideration at PLOS Pathogens. As with all papers reviewed by the journal, your manuscript was reviewed by members of the editorial board and by several independent reviewers. In light of the reviews (below this email), we would like to invite the resubmission of a significantly-revised version that takes into account the reviewers' comments.

We cannot make any decision about publication until we have seen the revised manuscript and your response to the reviewers' comments. Your revised manuscript is also likely to be sent to reviewers for further evaluation.

Sincerely,

Emily Speranza

Guest Editor

PLOS Pathogens

Michael Letko

Section Editor

PLOS Pathogens

Michael Malim

Editor-in-Chief

PLOS Pathogens

orcid.org/0000-0002-7699-2064

Reviewer's Responses to Questions

**Part I - Summary**

Reviewer #1: In this manuscript, Liukang et al describe gene expression in chickens infected with infectious bronchitis virus. IBV is understudied adding to the significance of this submission. This manuscript uses single cell sequencing to describe gene expression in the organs from infected animals. The study is comprehensive and the data are discussed in good detail. It is however, largely a descriptive study lacking mechanistic insight. It would be appropriate to consider the manuscript as a Resource article. Specific comments are below.

Reviewer #2: Liukang et al. present a detailed analysis employing immunohistochemistry and scRNA-seq to evaluate the target cells and inflammatory responses induced by infectious bronchitis virus (IBV) in chickens. The study has novelty in investigating a coronavirus of high importance to the poultry industry, and in the approaches employed, including detailed RNA-seq analyses of the Bursa of Fabricus and the chicken kidney which are novel. The study identifies unique target cells of infection in both tissues, though much was already known about infection of tracheal ciliated cells (not seen here) and infection renal tubular epithelial cells. The study identified acute inflammation in the distal nephrons with evidence for disrupted water-salt balance which could contribute to renal urate accumulation and mortality. Communication analysis also confirmed inflammatory signaling from infected cells in the distal nephron to immune cells. Overall this is a fairly thorough characterization of viral tropisms and the host response. The therapeutic implications for poultry are unclear, but this does represent an important piece of understanding animal and human health. However, there are some concerns, particularly with the analyses of infected cells, that limit the ability to draw biologically significant or novel conclusions. These concerns about that analytic approach significantly detract from this paper. Another limitation is that not all tissues were sampled, so an important host response could have been missed, even if it was not directly induced by infection.

**Part II – Major Issues: Key Experiments Required for Acceptance**

Reviewer #1: (No Response)

Reviewer #2: 1. It is unclear how the timepoint of 5 days post-infection was chosen, and somewhat unfortunate in that they were unable to detect any infection in the tracheal epithelium—presumably the first site of infection. Certainly a variety of time points may not be feasible but some justification and understanding would be helpful to orient the reader.

2. A major concern is that the detection of infected cells is not nearly as rigorous at is needs to be. For instance, in 2b the virus UTR is detected everywhere? Is it tropic for T cells. Clear demonstration of how many viral reads/cell and the background ambient amounts of viral UTR detection are needed. This is critical to evaluate the degree of background noise from ambient RNA amplification, etc. and should be controlled for in analyses. Algorithms such as Cellbender would allow the authors to control for ambient RNA detection and more definitively identify infected cells. The viral reads detected across the different cell types after controlling for this background detection should be provided. This is critical to any interpretation of the paper.

3. Figure 2f is uninterpretable despite trying to find a description in text, figure itself, legend, and methods. Is this cells from infected chickens compare to the same types from uninfected chickens? A volcano plot would make more sense? The same problem is noted in Figure. 7d

4. The authors state that they validated the RNA data using immunohistochemistry by noting colocalization between the N protein and the collecting duct marker AQP2 or distal tubule marker CALB1. However, in Fig. 2, it is unclear if there really is overlap between the collecting duct specific AQP2 and the IBV N protein as it looks to me they are proximal, but non-overlapping in panel a. There also appears to be some non-specific staining with the N-protein in both panels, but it’s odd that the IBV panel alone in panel b shows no staining whatsoever yet there is visible red staining in the overlay panel suggesting that the same exposure/filtering may not have been performed. Thus, it is hard to trust the conclusion that this corroborates the scRNA-seq data. The lack of data showing the infected cells also appears to be a lost opportunity to use the single cell data to explore how infected cells signal and communicate in comparison with uninfected cells. Such an analysis would significantly strengthen the manuscript.

5. Fig. 1F is highly unclear, as it is not stated what is being compared. Presumably this is infected relative to uninfected, but this is not explained. It would also be helpful to provide context in terms of how many genes are differentially expressed and what other genes might be different, as it’s unclear if these examples were cherry-picked.

6. In the bursa, the results of the scRNA-seq and IHC are diffuclt to reconcile. The authors detect viral RNA in macrophages, but cannot confirm this by RNA-seq. They also detect viral protein in FAE cells, but do not report whether these cells have RNA. The nomenclature between FAE cells and interfollicular epithelial cells is very confusing and needs to be clarified, with clear indication for which cells have both viral RNA and protein.

7. Several communication platforms also weight interactions by whether there is evidence for downstream signaling, and such an approach could be beneficial in finding ‘true’ receptor ligand interactions. This is important because just because a receptor and ligand are expressed does not necessarily mean that the interaction is active unless there is evidence of downstream signaling. It is unclear whether the signaling strength panels in Figs. 3e and 5e and 6g, it is unclear how the signaling strength of different pathways has been weighted. The figure legend needs more detail. Is this downstream signaling from these pathways or just the receptor/ligand? I could not find this description in the methods either.

8. Similarly in Fig. 5b, the co-localization between TIMD4 and N-protein is not apparent.

9. The layout and description of Figs. 7 and 8 is difficult to interpret. The full parameters for the WGCNA analysis are not described but are critical to interpretability and reproducibility. The rigor with which the modules are defined is not clear. For instance, a Jaccard index heatmap depicting the overlap in gene membership between all discovered modules would be more interpretable as in Kazer et al. PMID XX. The the expression of these modules across cell types and tissues could be displayed. Unclear what is meant by “enrichment of immune-related gene modules”. Rather that describing the modules by color, it might be nice to understand what these different modules described in terms of biology.

**Part III – Minor Issues: Editorial and Data Presentation Modifications**

Reviewer #1: 1. Line 36-STAT-This should probably be STAT1. If so, it is associated with more than apoptosis.

2. Line 77, line 601-1 day old chicks were used in this study. Justification for use of this age animals should be provided. Also more description of the IBV strain used in this study should be included. Not all strains infect the kidney, and chicks of this age commonly succumb to a respiratory infection, rather than renal disease. The infectious model should be clarified.

3. Line 180-“In conclusion our study demonstrates…”-This sentence should be modified since the data do not show a disruption of water-salt balance but rather are consistent with physiological studies showing this result.

4. Line 185-Much of the manuscript discusses cell-cell communication. These types of analyses should be discussed in more detail for those readers not familiar with the concept or approach.

5. Line 240-“Considering that bursa…”-Is there independent support for this hypothesis? If so, a reference should be provided.

6. Lines 259-267-Quantitative analyses, using Pearson’s coefficient should be provided to support these conclusions.

Reviewer #2: Small point. Figure 3d refers to panel a but should refer to panel c. Also text says 1b when it refers to 2b

PLOS authors have the option to publish the peer review history of their article (what does this mean?). If published, this will include your full peer review and any attached files.

Reviewer #1: No

Reviewer #2: No
---

## [Decision Letter · Decision Letter 1]

29 Apr 2024

Dear Prof. Zhang,

We are pleased to inform you that your manuscript 'Deciphering Infected Cell Types, Hub Gene Networks and Cell-Cell Communication in Infectious Bronchitis Virus via Single-Cell RNA Sequencing' has been provisionally accepted for publication in PLOS Pathogens.

Best regards,

Michael Letko, PhD

Section Editor

PLOS Pathogens

Michael Letko

Section Editor

PLOS Pathogens

Michael Malim

Editor-in-Chief

PLOS Pathogens

orcid.org/0000-0002-7699-2064

Reviewer Comments (if any, and for reference):

Reviewer's Responses to Questions

**Part I - Summary**

Reviewer #2: Liukang et al. revision review

Overall, Liukang et al. have addressed most of my comments, though mechanistic insight remains to be verified and agree that this manuscript may be more suitable as a resource.

**Part II – Major Issues: Key Experiments Required for Acceptance**

Reviewer #2: no major issues

**Part III – Minor Issues: Editorial and Data Presentation Modifications**

Reviewer #2: A few minor concerns should be addressed.

For clarity, in the abstract on line 35, it would be better to indicate that the transcripts for IL1B and IL18 are upregulated, because it is not clear whether the cytokine was measured directly

Line 38: should read “diverse cell types”

Lines 257-8. The statement is incorrect as CD40L is not a secreted protein, but it is expressed by CD4 T cells. This needs to be corrected.

PLOS authors have the option to publish the peer review history of their article (what does this mean?). If published, this will include your full peer review and any attached files.

Reviewer #2: No

---

## [Editor Report · Acceptance letter]

7 May 2024

Dear Prof. Zhang,

We are delighted to inform you that your manuscript, "Deciphering Infected Cell Types, Hub Gene Networks and Cell-Cell Communication in Infectious Bronchitis Virus via Single-Cell RNA Sequencing," has been formally accepted for publication in PLOS Pathogens.

Best regards,

Michael Malim

Editor-in-Chief

PLOS Pathogens

orcid.org/0000-0002-7699-2064